# EFFICIENT SPARSE PCA VIA BLOCK-DIAGONALIZATION

**Alberto Del Pia & Dekun Zhou**
University of Wisconsin-Madison
{delpia,dzhou44}@wisc.edu

**Yinglun Zhu**
University of California, Riverside
yzhu@ucr.edu

## ABSTRACT

Sparse Principal Component Analysis (Sparse PCA) is a pivotal tool in data analysis and dimensionality reduction. However, Sparse PCA is a challenging problem in both theory and practice: it is known to be NP-hard and current exact methods generally require exponential runtime. In this paper, we propose a novel framework to efficiently approximate Sparse PCA by (i) approximating the general input covariance matrix with a re-sorted block-diagonal matrix, (ii) solving the Sparse PCA sub-problem in each block, and (iii) reconstructing the solution to the original problem. Our framework is simple and powerful: it can leverage any off-the-shelf Sparse PCA algorithm and achieve significant computational speedups, with a minor additive error that is linear in the approximation error of the block-diagonal matrix. Suppose $g(k, d)$ is the runtime of an algorithm (approximately) solving Sparse PCA in dimension $d$ and with sparsity constant $k$. Our framework, when integrated with this algorithm, reduces the runtime to $\mathcal{O}\left(\frac{d}{d^\star} \cdot g(k, d^\star) + d^2\right)$, where $d^\star \leq d$ is the largest block size of the block-diagonal matrix. For instance, integrating our framework with the Branch-and-Bound algorithm reduces the complexity from $g(k, d) = \mathcal{O}(k^3 \cdot d^k)$ to $\mathcal{O}(k^3 \cdot d \cdot (d^\star)^{k-1})$, demonstrating exponential speedups if $d^\star$ is small. We perform large-scale evaluations on many real-world datasets: for exact Sparse PCA algorithm, our method achieves an average speedup factor of 100.50, while maintaining an average approximation error of 0.61%; for approximate Sparse PCA algorithm, our method achieves an average speedup factor of 6.00 and an average approximation error of -0.91%, meaning that our method oftentimes finds better solutions.

## 1 INTRODUCTION

In this paper, we study the Sparse Principal Component Analysis (Sparse PCA) problem, a variant of the well-known Principal Component Analysis (PCA) problem. Similar to PCA, Sparse PCA involves finding a linear combination of $d$ features that explains most variance. However, Sparse PCA distinguishes itself by requiring the use of only $k \ll d$ many features, thus integrating a sparsity constraint. This constraint significantly enhances interpretability, which is essential in data analysis when dealing with a large number of features. Sparse PCA has found widespread application across various domains, including text data analysis (Zhang & Ghaoui, 2011), cancer research (Hsu et al., 2014), bioinformatics (Ma & Dai, 2011), and neuroscience (Zhuang et al., 2020). For further reading, we refer interested readers to a comprehensive survey on Sparse PCA (Zou & Xue, 2018).

While being important and useful, Sparse PCA is a challenging problem—it is known to be NP-hard (Magdon-Ismail, 2017). Solving Sparse PCA exactly, such as through the Branch-and-Bound algorithm (Berk & Bertsimas, 2019), requires worst-case exponential runtime. Moreover, while there are near-optimal approximation algorithms that exhibit faster performance empirically, they still require worst-case exponential runtime (Bertsimas et al., 2022; Li & Xie, 2020; Cory-Wright & Pauphilet, 2022; Dey et al., 2022b; Zou et al., 2006; Dey et al., 2022a). This complexity impedes their applicability to large-scale datasets. On the other hand, there are numerous polynomial-time approximation algorithms for Sparse PCA (Chowdhury et al., 2020; Li & Xie, 2020; Papailiopoulos et al., 2013; Chan et al., 2015; Del Pia, 2022; Asteris et al., 2015). These algorithms typically face trade-offs between efficiency and solution quality. Some algorithms are fast but yield sub-optimal

solutions, others provide high-quality solutions at the cost of greater computational complexity, and a few achieve both efficiency and accuracy but only under specific statistical assumptions.

In this paper, we introduce a novel framework designed to efficiently approximate Sparse PCA through matrix block-diagonalization. Our framework facilitates the use of off-the-shelf algorithms in a plug-and-play manner. Specifically, the framework comprises three principal steps: (i) *Matrix Preparation:* Given a general input covariance matrix to Sparse PCA, we generate a re-sorted block-diagonal matrix approximation. This includes finding a denoised input matrix by thresholding, grouping non-zero entries into blocks, and slicing corresponding blocks in original input matrix. (ii) *Sub-Problem Solution:* Solve Sparse PCA sub-problems using any known algorithm in each block with a lower dimension, and find out the best solution among sub-problems; (iii) *Solution Reconstruction:* Reconstruct the best solution to the original Sparse PCA problem. We illustrate these steps in Fig. 1. At a high level, our approach involves solving several sub-problems in smaller matrices instead of directly addressing Sparse PCA in a large input matrix (Step (ii)), which significantly reduces computational runtime. This is made possible by carefully constructing sub-problems (Step (i)) and efficiently reconstructing an approximate solution to the original problem (Step (iii)).

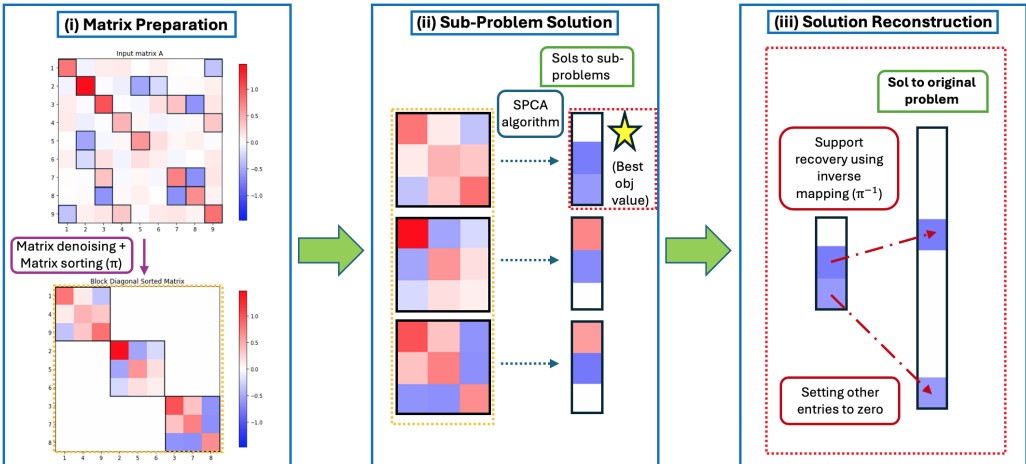

Figure 1: Illustration of our proposed approach, given a $9 \times 9$ covariance input matrix $A$. (i) Entries away from zero are highlighted in the upper matrix (original input $A$). Then group those entries in blocks, zero out outside entries, sort the matrix, and obtain the lower block-diagonal approximation. Heatmaps present values of matrix entries. The axes are indices of $A$; (ii) Extract sub-matrices from the block-diagonal approximation, and solve sub-problems via a suitable Sparse PCA algorithm; (iii) Select the solution with the highest objective value obtained from the sub-problems. Construct a solution for the original Sparse PCA problem by mapping its non-zero entries to their original locations using the inverse mapping of the sorting process, and setting all other entries to zero.

Our framework significantly speeds up the computation of Sparse PCA, with negligible approximation error. We theoretically quantify the computational speedup and approximation error of our framework in Sections 3 and 4, and conduct large scale empirical evaluations in Section 5. Next, we illustrate the performance of our method via a concrete example. Suppose one is given an input covariance matrix being a noisy sorted block-diagonal matrix with 10 blocks, each of size 100, and is asked to solve Sparse PCA with a sparsity constant 4. The Branch-and-Bound algorithm (Berk & Bertsimas, 2019) takes over 3600 seconds and obtains a solution with objective value 13.649. Our framework integrated with Branch-and-Bound algorithm reduces runtime to 17 seconds, a speedup by a factor of 211, and obtains a solution with objective value 13.637, with an approximation error of 0.09%.

**Our contributions.** We summarize our contributions in the following:

- We propose a novel framework for approximately solving Sparse PCA via matrix block-diagonalization. This framework allows users to reuse any known algorithm designed for Sparse PCA in the literature in a very easy way, i.e., simply applying the known algorithm to the blocks of the approximate matrix, and accelerates the computation. To the best of our knowledge, our work is the first to apply block-diagonalization to solving Sparse PCA problem.

- We provide approximation guarantees for our framework. We show that when integrated with an approximation algorithm, our method can generally obtain a better multiplicative factor, with an additional cost of an additive error that is linear in the approximation error of the block-diagonal approximate matrix. Under certain assumptions, the additive error can also be improved.
- We also perform a time complexity analysis showing that for an (approximation) algorithm with runtime $g(k, d)$, our framework could reduce the runtime to $\mathcal{O}(\frac{d}{d^\star} \cdot g(k, d^\star) + d^2)$, where $d$ is the dimension of Sparse PCA input covariance matrix, $k$ is the sparsity constant, and $d^\star$ is the largest block size in the block-diagonal approximate matrix.
- We show that for a statistical model that is widely studied in robust statistics, there exists an efficient way to find a block-diagonal approximation matrix. Moreover, we provide a very simple way to extend to the setting where the statistical assumptions are dropped and where computational resources are taken into consideration.
- We conduct extensive experiments to evaluate our framework, and our empirical study shows that (i) when integrated with Branch-and-Bound algorithm, our framework achieves an average speedup factor of 100.50, while maintaining an average approximation error of 0.61%, and (ii) when integrated with Chan's algorithm (Chan et al., 2015), our framework achieves an average speedup factor of 6.00 and an average approximation error of -0.91%, meaning that our method oftentimes finds better solutions.

**Our techniques.** In this paper, we also introduce several novel techniques that may be of independent interest. We characterize a structural property of an optimal solution to Sparse PCA, when given a block-diagonal input matrix. We establish that solving Sparse PCA for a block-diagonal matrix input is equivalent to addressing individual Sparse PCA sub-problems within blocks, which consequently provides significant computational speedups. This result is non-trivial: while the support of optimal solutions might span multiple blocks, we prove that there always exists an optimal solution whose support is contained within a single block, ensuring the efficiency of our framework. Additionally, leveraging the defined structure, we propose a novel binary-search based method to identify the best block-diagonal approximation, particularly when computational resources are limited.

**Paper organization.** This paper is organized as follows. In Section 2, we introduce Sparse PCA problem formally, and provide motivation of our proposed framework. In Section 3, we provide details of operational procedures of our proposed framework, and develop guarantees of approximation errors as well as runtime complexity. In Section 4 we further extend our algorithms to learning problems either under (i) a statistical model, or under (ii) a model-free setting, and provide theoretical guarantees respectively. In Section 5, we run large-scale empirical evaluations and demonstrate the efficacy of our proposed framework when integrated with both exact and approximate Sparse PCA algorithms. We defer discussion of additional related work, including existing block-diagonalization methods in the literature, detailed proofs, and additional empirical results to the appendix.

## 2 PROBLEM SETTING AND MOTIVATION

In this section, we define formally the problem of interest in this paper, and then we explain the motivation behind our framework. We first define the Sparse PCA problem as follows:

$$\mathsf{OPT} := \max \ x^\top A x \quad \text{s.t.} \ \|x\|_2 = 1, \ \|x\|_0 \leq k. \tag{SPCA}$$

Here, a symmetric and positive semidefinite input matrix $A \in \mathbb{R}^{d \times d}$, and a positive integer $k \leq d$ are given. $k$ is known as the sparsity constant, the upper bound on the number of nonzero entries in solution $x$.

**Motivating idea in this paper.** The motivation for our method is illustrated through the following example, which demonstrates the efficiency of solving sub-problems: Suppose that a user is given an input block-diagonal covariance matrix $A$ of size $d$, comprising $d/d^\star$ many $d^\star \times d^\star$ blocks (assume that $d$ is divisible by $d^\star$), and is asked to solve SPCA with sparsity constant $k$. The user can simply apply an exact algorithm (e.g., Branch-and-Bound algorithm) to solve the large problem in time $\mathcal{O}(k^3 \cdot \binom{d}{k}) = \mathcal{O}(k^3 \cdot d^k)$. The user could also solve $d/d^\star$ many sub-problems in each block, potentially obtaining an optimal solution (a fact we rigidly prove in Section 3.2.1), in time $\mathcal{O}((d/d^\star) \cdot k^3 \cdot (d^\star)^k)$. This segmented approach yields a significant speedup factor of $\mathcal{O}((d/d^\star)^{k-1})$.

However, not all input matrices $A$ are of the block-diagonal structure, even after permuting its rows and columns. One goal of our work is to develop a reasonable way to construct a (sorted)

block-diagonal matrix approximation $\widetilde{A}$ to $A$ such that they are close enough. In this paper, we are interested in finding the following $\varepsilon$-*matrix approximation* to $A$:

**Definition 1** ($\varepsilon$-matrix approximation). *Given a matrix $A \in \mathbb{R}^{d \times d}$, denote by $A_{ij}$ the $(i,j)$-th entry of $A$. An $\varepsilon$-matrix approximation of $A$ is defined as a $d \times d$ matrix $\widetilde{A}$ such that*

$$\left\| A - \widetilde{A} \right\|_\infty := \max_{i,j \in [d]} \left| A_{ij} - \widetilde{A}_{ij} \right| \leq \varepsilon.$$

*Moreover, we define the set of all $\varepsilon$-matrix approximations of $A$ to be $\mathcal{B}(A, \varepsilon)$.*

In other words, $\widetilde{A} \in \mathcal{B}(A, \varepsilon)$ if and only if $\widetilde{A}$ belongs to an $\ell_\infty$-ball centered at $A$ with radius $\varepsilon$. Our overarching strategy is that if the input matrix is not block-diagonal, we find a matrix approximation in the ball so that it could be resorted to be block-diagonal. We then solve the Sparse PCA problem using this resorted matrix and reconstruct the approximate solution to the original problem, taking advantage of the computational speedups via addressing smaller sub-problems.

**Additional notation.** We adopt non-asymptotic big-oh notation: For functions $f, g : \mathcal{Z} \rightarrow \mathbb{R}_+$, we write $f = \mathcal{O}(g)$ (resp. $f = \Omega(g)$) if there exists a constant $C > 0$ such that $f(z) \leq Cg(z)$ (resp. $f(z) \geq Cg(z)$) for all $z \in \mathcal{Z}$. For an integer $n \in \mathbb{N}$, we let $[n]$ denote the set $\{1, 2, \ldots, n\}$. For a vector $z \in \mathbb{R}^d$, we use $z_i$ to denote its $i$-th entry. We denote $\text{supp}(z) := \{i \in [d] : z_i \neq 0\}$ the *support* of $z$. For $1 \leq q \leq \infty$, we denote $\|z\|_q$ the $q$-norm of $z$, i.e., $\|z\|_q := (\sum_{i=1}^d |z_i|^q)^{1/q}$, and $\|z\|_\infty := \max_{i \in [d]} |z_i|$. For a matrix $A \in \mathbb{R}^{n \times m}$, we use $A_{ij}$ to denote its $(i, j)$-th entry. We define $\|A\|_\infty := \max_{i,j} |A_{ij}|$. For index sets $S \subseteq [n]$, $T \subseteq [m]$, we denote $A_{S,T}$ the $|S| \times |T|$ sub-matrix of $A$ with row index $S$ and column index $T$. For matrices $A_i \in \mathbb{R}^{d_i \times d_i}$ for $i = 1, 2, \ldots, p$, we denote $\text{diag}(A_1, A_2, \ldots, A_p)$ the $(\sum_{i=1}^p d_i) \times (\sum_{i=1}^p d_i)$ block-diagonal matrix with blocks $A_1, A_2, \ldots, A_p$. For a square matrix $B \in \mathbb{R}^{d \times d}$, we define the *size* of $B$ to be $d$.

## 3 OUR APPROACH

In this section, we introduce our framework for solving SPCA approximately via block-diagonalization technique. We explain in detail the operational procedures of our framework in Section 3.1. In Section 3.2, we characterize the approximation guarantees and computational speedups of our framework. Note that, in this section, our proposed procedures require a predefined threshold $\varepsilon > 0$. In Section 4, we extend our algorithms to learning the threshold in a statistical model, as well as in a model-free setting.

### 3.1 ALGORITHMS

In this section, we outline the operational procedure of our proposed framework for solving SPCA approximately. The procedure commences with the input matrix $A \in \mathbb{R}^{d \times d}$ and a predefined threshold value $\varepsilon > 0$. Initially, a denoised matrix $A^\varepsilon$ is derived through thresholding as described in Algorithm 1. Subsequently, several sub-matrices and their corresponding sets of indices are extracted from $A$ and $A^\varepsilon$ by employing Algorithm 2, which groups index pair $(i, j)$ if $A_{ij}^\varepsilon \neq 0$. This grouping mechanism can be efficiently implemented using a Depth-First Search approach, which iterates over each unvisited index $i$ and examines all non-zero $A_{ij}^\varepsilon$, and then visit each corresponding index $j$ that is unvisited. The computational complexity of Algorithm 2 is bounded by $\mathcal{O}(d^2)$. Finally, an approximate solution to SPCA is obtained by solving several Sparse PCA sub-problems in each block using a certain (approximation) algorithm $\mathcal{A}$, mapping the solution back to the original problem, and finding out the best solution. The operational details are further elaborated in Algorithm 3.[1]

### 3.2 APPROXIMATION GUARANTEES AND COMPUTATIONAL SPEEDUPS

In this section, we provide analysis of approximation guarantees and speedups of our framework Algorithm 3. Before showing approximation guarantees, we first define the $\varepsilon$-*intrinsic dimension of $A$*:

---

[1] We note that on line 6 in Algorithm 3, if $k$ is larger than the size of $\widetilde{A}_i$, a PCA problem is solved instead.

---

**Algorithm 1** Denoising procedure via thresholding

---

1: **Input:** Matrix $A \in \mathbb{R}^{d \times d}$, threshold $\varepsilon > 0$
2: **Output:** Denoised matrix $A^\varepsilon \in \mathbb{R}^{d \times d}$
3: **for** $(i, j) = (1, 1)$ **to** $(d, d)$ **do**
4:     If $|A_{ij}| > \varepsilon$, set $A_{ij}^\varepsilon \leftarrow A_{ij}$, otherwise set $A_{ij}^\varepsilon \leftarrow 0$
5: **Return** $A^\varepsilon$

---

**Algorithm 2** Matrix block-diagonlization

---

1: **Input:** Matrices $A, A^\varepsilon \in \mathbb{R}^{d \times d}$
2: **Output:** Sub-matrices and their corresponding lists of indices
3: **for** each pair of indices $(i, j) \in [d] \times [d]$ **do**
4:     Group $i$ and $j$ together if $|A_{ij}^\varepsilon| > 0$
5: Obtain sets of indices $\{S_i\}_{i=1}^p$ that are grouped together
6: **Return** sub-matrices $A_{S_i, S_i}$ and the corresponding $S_i$, for $i \in [p]$

---

**Definition 2** ($\varepsilon$-intrinsic dimension). *Let $A \in \mathbb{R}^{d \times d}$. The $\varepsilon$-intrinsic dimension of A, denoted by* $\mathrm{int\,dim}(A, \varepsilon)$, *is defined as:*

$$\mathrm{int\,dim}(A, \varepsilon) := \min_{B \in \mathcal{B}(A, \varepsilon)} \mathrm{lbs}(B),$$

*where $\mathrm{lbs}(B)$ denotes the largest block size of B, i.e., the largest among $\{|S_i|\}_{i \in [p]}$ after running lines 3-5 in Algorithm 2 with $A^\varepsilon = B$.*

As we will see later in this section, this concept plays a crucial role in characterizing both the quality of the approximate solution obtained through our framework, Algorithm 3, and the computational speedup achieved by our framework. Moreover, a matrix $\widetilde{A}$ can be efficiently identified such that $\mathrm{lbs}(\widetilde{A}) = \mathrm{int\,dim}(A, \varepsilon)$ using Algorithm 1, as shown in Lemma 1.

**Lemma 1.** *Let $A \in \mathbb{R}^{d \times d}$ and $\varepsilon > 0$. Given input $(A, \varepsilon)$, Algorithm 1 outputs an $\varepsilon$-approximation of A, denoted as $\widetilde{A}$, such that $\mathrm{lbs}(\widetilde{A}) = \mathrm{int\,dim}(A, \varepsilon)$ in time $\mathcal{O}(d^2)$.*

### 3.2.1 APPROXIMATION GUARANTEES

In this section, we provide approximation guarantees for our framework Algorithm 3. We first show that in Theorem 1 that, an optimal solution to SPCA with input $(A, k)$ and an optimal solution to $(A^\varepsilon, k)$ are close to each other in terms of objective value:

**Theorem 1.** *Let $A \in \mathbb{R}^{d \times d}$ be symmetric, $k \leq d$, and $\varepsilon > 0$. Denote by $A^\varepsilon$ the output of Algorithm 1 with input $(A, \varepsilon)$, $\widetilde{x}$ an optimal solution to SPCA with input $(A^\varepsilon, k)$, and $x^\star$ an optimal solution to SPCA with input $(A, k)$. Then, it follows that*

$$\left| (x^\star)^\top A x^\star - \widetilde{x}^\top A \widetilde{x} \right| \leq 2k \cdot \varepsilon.$$

---

**Algorithm 3** Efficient Sparse PCA via block-diagonalization

---

1: **Input:** Matrix $A \in \mathbb{R}^{d \times d}$, positive integer $k \leq d$, threshold $\varepsilon > 0$, algorithm $\mathcal{A}$ for (approximately) solving SPCA
2: **Output:** Approximate solution to SPCA
3: Obtain denoised matrix $A^\varepsilon$ using Algorithm 1 with input $(A, \varepsilon)$
4: Obtain sub-matrices $\{\widetilde{A}_i\}_{i=1}^p$ and corresponding index sets $\{S_i\}_{i=1}^p$ via Algorithm 2 with $(A, A^\varepsilon)$
5: **for** $i = 1$ **to** $p$ **do**
6:     Solve SPCA approximately with input $(\widetilde{A}_i, k)$ using algorithm $\mathcal{A}$ and obtain solution $x_i$
7:     Construct $y_i \in \mathbb{R}^d$ by placing $(x_i)_j$ at $(y_i)_{S_i(j)}$ for $j$ in block indices $S_i$ and zero elsewhere
8: **Return** the best solution among $\{y_i\}_{i=1}^p$ that gives the largest $y_i^\top A y_i$

---

Then, in Theorem 2, we show that solving SPCA exactly with a block-diagonal input matrix $\widetilde{A}$ is equivalent to addressing SPCA sub-problems in blocks within $\widetilde{A}$:

**Theorem 2.** *Let $\widetilde{A} = \mathsf{diag}(\widetilde{A}_1, \widetilde{A}_2, \ldots, \widetilde{A}_p)$ be a symmetric block-diagonal matrix. Denote* OPT *to be the optimal value to SPCA with input pair $(\widetilde{A}, k)$. Let* $\mathsf{OPT}_i$ *to be the optimal value to SPCA with input pair $(\widetilde{A}_i, k)$, for $i \in [p]$. Then, one has* $\mathsf{OPT} = \max_{i \in [p]} \mathsf{OPT}_i$.

**Remark 1.** *Theorem 2 is particularly non-trivial: without this result, one might think the support of any optimal solution to SPCA may span different blocks. Solving SPCA with a block-diagonal input would consequently involve iterating over all possible sparsity constants (that sum up to $k$) for each block-diagonal sub-matrix, making our framework proposed in Section 3.1 nearly impractical.*

We now extend Theorem 1 to incorporate the use of various approximation algorithms. We note that our extension improves the general bound proposed in Theorem 1, since Algorithm 3 finds a specific block-diagonal approximation of $A$. Furthermore, Theorem 3 suggests that our framework Algorithm 3, can also be highly effective when combined with other approximation algorithms. Specifically, it has the potential to find a better solution, particularly when the value of $\varepsilon$ is small.

**Theorem 3.** *Let $A \in \mathbb{R}^{d \times d}$ be symmetric and positive semidefinite, let $k$ be a positive integer, and denote by $x^\star \in \mathbb{R}^d$ an optimal solution to SPCA with input $(A, k)$. Suppose that an algorithm $\mathcal{A}$ for SPCA with input $(A, k)$ finds an approximate solution $x \in \mathbb{R}^d$ to SPCA with multiplicative factor $m(k, d) \geq 1$ and additive error $a(k, d) \geq 0$, i.e., one has $x^\top Ax \geq (x^\star)^\top Ax^\star / m(k, d) - a(k, d)$. Furthermore, we assume that $m(k, d)$ and $a(k, d)$ is non-decreasing with respect to $d$. For $\varepsilon > 0$, denote by $y \in \mathbb{R}^d$ be the output of Algorithm 3 with input tuple $(A, k, \varepsilon, \mathcal{A})$. Then, one has*

$$y^\top Ay \geq \frac{(x^\star)^\top Ax^\star}{m\left(k, \mathrm{int}\, \dim(A, \varepsilon)\right)} - a\left(k, \mathrm{int}\, \dim(A, \varepsilon)\right) - \frac{1}{m\left(k, \mathrm{int}\, \dim(A, \varepsilon)\right)} \cdot k\varepsilon.$$

**Remark 2.** *Denote by $d^\star := \mathrm{int}\, \dim(A, \varepsilon) \leq d$. Theorem 3 implies that our framework, Algorithm 3, when inputted with $(A, k, \varepsilon, \mathcal{A})$, could find an approximate solution with a better multiplicative factor $m(k, d^\star)$, and an additive error $a(k, d^\star) + \frac{k\varepsilon}{m(k, d^\star)}$. Note that the additive error is also improved if*

$$a(k, d) \geq a\left(k, \mathrm{int}\, \dim(A, \varepsilon)\right) + \frac{1}{m\left(k, \mathrm{int}\, \dim(A, \varepsilon)\right)} \cdot k\varepsilon.$$

*In addition, we have found instances where our framework can indeed obtain better solutions when integrated with some approximation algorithm, as shown in Table 9 in Appendix C. Finally, we summarized in Table 1 the change of multiplicative factor and additive error when integrated our framework to some existing (approximation) algorithms for SPCA.*

Table 1: Comparison of multiplicative factors (MF) and additive errors (AE) when integrating our framework with different algorithms. The MISDP and Greedy algorithms are both studied in Li & Xie (2020). Note that our MF is always no larger than the original MF.

| Algorithm | MF | AE | Our MF | Our AE |
|---|---|---|---|---|
| Chan et al. (2015) | $\min(\sqrt{k}, d^{1/3})$ | $0$ | $\min(\sqrt{k}, \mathrm{int}\, \dim(A, \varepsilon)^{1/3})$ | $k\varepsilon/$Our MF |
| MISDP | $\min(k, d/k)$ | $0$ | $\min(k, \mathrm{int}\, \dim(A, \varepsilon)/k)$ | $k\varepsilon/$Our MF |
| Greedy | $k$ | $0$ | $k$ | $\varepsilon$ |
| Exact algorithms | $1$ | $0$ | $1$ | $k\varepsilon$ |
| Asteris et al. (2015) | $1$ | $\delta$ | $1$ | $k\varepsilon + \delta$ |

### 3.2.2 COMPUTATIONAL SPEEDUPS

In this section, we characterize the runtime complexity of Algorithm 3. We will see that the $\varepsilon$-intrinsic dimension also plays an important role in this section.

**Proposition 1.** *Let $A \in \mathbb{R}^{d \times d}$, and let $k$ be a positive integer. Suppose that for a specific algorithm $\mathcal{A}$, the time complexity of using $\mathcal{A}$ for (approximately) solving SPCA with input $(A, k)$ is a function $g(k, d)$ which is convex and non-decreasing with respect to $d$. We assume that $g(k, 1) = 1$. Then, for a given threshold $\varepsilon > 0$, the runtime of Algorithm 3 with input tuples $(A, k, \varepsilon, \mathcal{A})$ is at most*

$$\mathcal{O}\left(\left\lceil \frac{d}{\mathrm{int}\, \dim(A, \varepsilon)} \right\rceil \cdot g\left(k, \mathrm{int}\, \dim(A, \varepsilon)\right) + d^2\right)$$

**Remark 3.** *We remark that the assumption $g(k, d)$ is convex and non-decreasing with respect to $d$ in Proposition 1 is a very weak assumption. Common examples include $d^\alpha$ and $d^\alpha \log d$ with $\alpha \geq 1$.*

**Remark 4.** *In this remark, we explore the computational speedups provided by Algorithm 3 by modeling $\text{int dim}(A, \varepsilon)$ as a function of $A$ and $\varepsilon$. We assume without loss of generality that $\|A\|_\infty = 1$, thereby setting $\text{int dim}(A, 1) = 0$. It is important to note that while $\text{int dim}(A, \varepsilon)$ is discontinuous, for the purpose of illustration in Table 2, it is treated as a continuous function of $\varepsilon$.*

Table 2: Comparison of original runtimes and current runtimes in our framework. We assume that the algorithm has runtime $\mathcal{O}(d^\alpha)$, for some $\alpha \geq 2$.

| $\text{int dim}(A, \varepsilon)$ | Runtime | Our Runtime | Speedup factor |
|---|---|---|---|
| $d(1 - e^{-c(1-\varepsilon)})$ | $\mathcal{O}(d^\alpha)$ | $\mathcal{O}(d^\alpha(1 - e^{-c(1-\varepsilon)})^{\alpha-1})$ | $\mathcal{O}((1 - e^{-c(1-\varepsilon)})^{1-\alpha})$ |
| $d(1 - \varepsilon)$ | $\mathcal{O}(d^\alpha)$ | $\mathcal{O}(d^\alpha(1 - \varepsilon)^{\alpha-1})$ | $\mathcal{O}((1 - \varepsilon)^{1-\alpha})$ |
| $d(1 - \varepsilon)^m$ | $\mathcal{O}(d^\alpha)$ | $\mathcal{O}(d^\alpha(1 - \varepsilon)^{(\alpha-1)m})$ | $\mathcal{O}((1 - \varepsilon)^{(1-\alpha)m})$ |

## 4 EXTENSIONS

Our algorithm presented in Section 3.1 takes a known threshold as an input. In this section, we remove this constraint by extending our algorithm to various settings. Specifically, in Section 4.1, we consider learning under a statistical model, where one can do the denoising of a matrix easily. In Section 4.2, we propose a practical learning approach without any assumption.

### 4.1 LEARNING UNDER A STATISTICAL MODEL

A crucial aspect of our proposed framework involves determining an appropriate threshold, $\varepsilon$, as required by Algorithm 3. While specifying such a threshold is generally necessary, there are many instances where it can be effectively estimated from the data. In this section, we discuss a model commonly employed in robust statistics (Comminges et al., 2021; Kotekal & Gao, 2023; 2024). This model finds broad applications across various fields, including clustering (Chen & Witten, 2022) and sparse linear regression (Minsker et al., 2022).

Before presenting the formal definition, we first establish some foundational intuition about the model. Conceptually, if the input data matrix $A$ is viewed as an empirical covariance matrix, and based on the intuition that, a feature is expected to exhibit strong correlations with only a few other features, it follows that $A$ should be, or at least be close to, a block-diagonal matrix. This observation motivates us for considering Model 1:

**Model 1.** *The input matrix $A \in \mathbb{R}^{d \times d}$ is close to a block-diagonal symmetric matrix $\widetilde{A} \in \mathbb{R}^{d \times d}$, i.e., $A = \widetilde{A} + E$ for some symmetric matrix $E \in \mathbb{R}^{d \times d}$ such that $A$ is positive semidefinite and*

*(i) for $i \leq j$, each entry $E_{ij}$ is drawn from an i.i.d. $\varrho$-sub-Gaussian distribution, with mean zero, and variance $\sigma^2$, where $\varrho > 0$ and $\sigma^2$ being unknown, but an upper bound $u \geq \varrho^2/\sigma^2$ is given;*

*(ii) write $\widetilde{A} = \text{diag}\left(\widetilde{A}_1, \widetilde{A}_2, \ldots, \widetilde{A}_p\right)$, the size of each diagonal block $\widetilde{A}_i$ is upper bounded by $d^\star$, where $d^\star \leq Cd^\alpha$ for some known constant $C > 0$ and $\alpha \in (0, 1)$, but $d^\star$ is unknown.*

In assumption (i) of Model 1, the sub-Gaussian distribution is particularly favored as this class of distributions is sufficiently broad to encompass not only the Gaussian distribution but also distributions of bounded variables, which are commonly observed in noise components of real-world datasets. Assumption (ii) of Model 1 uses the constant $\alpha$ to quantify the extent to which features are clustered.

The procedure for estimating $\|E\|_\infty$ in Model 1 involves an estimation of the variance $\sigma^2$, which is adapted from Comminges et al. (2021), and then multiply a constant multiple of $\sqrt{\log d}$. We present the details in Algorithm 4:

In the next theorem, we provide a high probability approximation bound, as well as a runtime analysis, for Algorithm 3 using the threshold $\bar{\varepsilon}$ found via Algorithm 4. The results imply that in Model 1, one can efficiently solve SPCA by exploiting the hidden sparse structure.

**Proposition 2.** *Consider Model 1, and denote by $\bar{\varepsilon}$ the output of Algorithm 4 with input tuple $(A, C, \alpha, u)$. Let $k \leq d$ be a positive integer, and assume that $d$ satisfies that $d^{1-\alpha} > C_0 \cdot (C + 1) \cdot$*

---

**Algorithm 4** Estimation of $\|E\|_\infty$ in Model 1

---

1: **Input:** Matrix $A \in \mathbb{R}^{d \times d}$, parameters $C > 0$, $\alpha \in (0,1)$, and $u$ in Model 1
2: **Output:** A number $\bar{\varepsilon} > 0$ as an estimate of $\|E\|_\infty$
3: Divide the set of indices $\{(i,j) \in [d] \times [d] : i \leq j\}$ into $m = \lfloor (2C+2)d^{1+\alpha} \rfloor$ disjoint subsets
    $B_1, B_2, \ldots, B_m$ randomly, ensuring each subset $B_i$ has cardinality lower bounded by $\left\lfloor \frac{d^2+d}{2m} \right\rfloor$
4: Initialize an array $S$ to store subset variances
5: **for** $i = 1$ **to** $m$ **do**
6:     $S_i \leftarrow \frac{1}{|B_i|} \sum_{(i,j) \in B_i} A_{ij}^2$
7: $\bar{\sigma}^2 \leftarrow \text{median}(S)$
8: **Return** $\bar{\varepsilon} \leftarrow 2u\bar{\sigma}\sqrt{3 \log d}$

---

$u \log(8C + 8)$ *for some large enough absolute constant* $C_0 > 0$. *Then, the following holds with probability at least* $1 - 2d^{-1} - \exp\{2 \log d - d^{1+\alpha}/(4C + 4)\}$:

*(i) Denote by $\widetilde{x}^\star$ the optimal solution to SPCA with input $(\widetilde{A}, k)$. For an (approximation) algorithm $\mathcal{A}$ with multiplicative factor $m(k,d) \geq 1$ and additive error $a(k,d) \geq 0$, where the functions $m$ and $a$ is non-decreasing with respect to $d$, the output $y$ of Algorithm 3 with input tuple $(A, k, \bar{\varepsilon}, \mathcal{A})$ satisfies that*

$$y^\top \widetilde{A} y \geq \frac{(\widetilde{x}^\star)^\top \widetilde{A} \widetilde{x}^\star}{m(k, d^\star)} - a(k, d^\star) - \left(1 + \frac{2}{m(k, d^\star)}\right) \cdot k\bar{\varepsilon}.$$

*(ii) If, in addition, the time complexity of using $\mathcal{A}$ for (approximately) solving SPCA with input $(A, k)$ is a function $g(k, d)$ which is convex and non-decreasing with respect to $d$, and satisfies $g(k, 1) = 1$. Then, the runtime of Algorithm 3 with input tuples $(A, k, \varepsilon, \mathcal{A})$ is at most*

$$\mathcal{O}\left(\left\lceil \frac{d}{d^\star} \right\rceil \cdot g(k, d^\star) + d^2\right)$$

Finally, we remark that, the assumption $d^{1-\alpha} > C_0 \cdot (C + 1) \cdot u \log(8C + 8)$ is true when $d$ is large enough, as $C_0$, $C$, and $u$ are usually some constants not changing with respect to the dimension $d$.

## 4.2 A PRACTICAL LEARNING APPROACH

In this section, we describe a practical algorithm to estimate $\varepsilon$ without prior statistical assumptions while simultaneously providing an approximate solution to SPCA.

Recall from Algorithm 3 that for a given threshold $\varepsilon$, a thresholded matrix $A^\varepsilon$ is generated via Algorithm 1 from the original matrix $A$. This matrix $A^\varepsilon$ is then utilized to approximate SPCA using a specified algorithm $\mathcal{A}$. Our proposed approach in this section involves initially selecting an $\varepsilon$ value within a predefined range, executing Algorithm 3 with this $\varepsilon$, and subsequently adjusting $\varepsilon$ in the following iterations through a binary search process. Since efficiency is of our major interest; thus, we ask users to provide a parameter $d_0$ that indicates the maximum dimension they are willing to compute due to computational resource limit. We defer the detailed description of this binary-search based algorithm to Algorithm 5. For clarity, let us denote by $\mathsf{OPT}(\varepsilon)$ the objective value of $(x^\varepsilon)^\top A x^\varepsilon$, where $x^\varepsilon$ is the output of Algorithm 3 with inputs $(A, k, \varepsilon, \mathcal{A})$.

We have the characterization of approximation error and runtime complexity of Algorithm 5:

**Theorem 4.** *Let $A \in \mathbb{R}^{d \times d}$ be symmetric and positive semidefinite, and let $k$ be a positive integer. Suppose that an algorithm $\mathcal{A}$ for SPCA with input $(A, k)$ finds an approximate solution $x \in \mathbb{R}^d$ to SPCA such that has multiplicative factor $m(k, d) \geq 1$ and additive error $a(k, d) \geq 0$, with $m(k, d)$ and $a(k, d)$ being non-decreasing with respect to $d$. Suppose that Algorithm 5 with input $(A, k, \delta, \mathcal{A}, d_0)$ terminates with $\mathsf{OPT}(\varepsilon^\star) \geq 0$. Then one has*

$$\mathsf{OPT}(\varepsilon^\star) \geq \frac{1}{m(k, d_0)} \cdot \mathsf{OPT}(0) - a(k, d_0) - \frac{1}{m(k, d_0)} \cdot k\varepsilon^\star.$$

*Suppose that for $\mathcal{A}$, the time complexity of using $\mathcal{A}$ for (approximately) solving SPCA with input $(A, k)$ is a function $g(k, d)$ which is convex and non-decreasing with respect to $d$, $g(k, 1) = 1$, and*

---

**Algorithm 5** A practical version of Algorithm 3 without a given threshold

---

1: **Input:** Matrix $A \in \mathbb{R}^{d \times d}$, positive integer $k \leq d$, tolerance parameter $\delta$, (approximate) algorithm $\mathcal{A}$ for SPCA, and integer $d_0$
2: **Result:** Approximate solution to Problem SPCA
3: $L \leftarrow 0, U \leftarrow \|A\|_\infty$, find $\mathsf{OPT}(U)$ via $\mathcal{A}$
4: **while** $U - L > \delta$ **do**
5:     $\varepsilon \leftarrow (L + U)/2$
6:     Find $A^\varepsilon$ via Algorithm 1, and the largest block size $d^\star := \mathrm{lbs}(A^\varepsilon)$
7:     **if** problems with largest block size $\geq d^\star$ has been solved before **then** $U \leftarrow \varepsilon$; **continue**
8:     **if** $d^\star > d_0$ **then** $L \leftarrow \varepsilon$; **continue**
9:     Find $\mathsf{OPT}(\varepsilon)$ via Algorithm 3 with algorithm $\mathcal{A}$
10:     **if** $\mathsf{OPT}(\varepsilon)$ does not improve **then** $U \leftarrow \varepsilon$
11: **return** best $\mathsf{OPT}(\varepsilon)$ found and the corresponding $x^\varepsilon$

---

$g(k, 0) = 0$. *The runtime for Algorithm 5 is at most*

$$\mathcal{O}\left(\log\left(\frac{\|A\|_\infty}{\delta}\right) \cdot \left(\left\lceil \frac{d}{d_0} \right\rceil \cdot g\left(k, d_0\right) + d^2\right)\right)$$

**Remark 5.** *In this remark, we discuss the practical advantages of Algorithm 5:*

- *The first direct advantage is that Algorithm 5 does not rely on any assumptions, enabling users to apply the framework to any dataset. Moreover, the framework can be applied to scenarios where an improved estimation of the binary search range is known. For instance, if users possess prior knowledge about an appropriate threshold $\bar{\varepsilon}$ for binary search, they can refine the search interval in Algorithm 5 by setting the lower and upper bounds to $L := a\bar{\varepsilon}$ and $U := b\bar{\varepsilon}$ for some proper $0 \leq a < b$ to improve the efficiency by a speedup factor of $\mathcal{O}(\log(\|A\|_\infty / \delta) / \log((b - a)\bar{\varepsilon}/\delta))$.*
- *Secondly, as detailed in Theorem 4, is that Algorithm 5 achieves a similar trade-off between computational efficiency and approximation error as Algorithm 1, which are proposed in Theorem 3 and Proposition 1. The time complexity of Algorithm 5, compared to Algorithm 3, increases only by a logarithmic factor due to the binary search step, thus preserving its efficiency.*
- *The third advantage is that this algorithm considers computational resources, ensuring that each invocation of Algorithm 3 remains efficient.*

## 5   EMPIRICAL RESULTS

In this section, we report the numerical performance of our proposed framework.

**Datasets.** We perform numerical tests in datasets widely studied in feature selection, including datasets CovColon from Alon et al. (1999), LymphomaCov1 from Alizadeh et al. (1998), Reddit1500 and Reddit2000 from Dey et al. (2022b), LeukemiaCov, LymphomaCov2, and ProstateCov from Dettling (2004), ArceneCov and DorotheaCov from Guyon et al. (2007), and GLI85Cov and GLABRA180Cov from Li (2020). We take the sample covariance matrices of features in these datasets as our input matrices $A$. The dimensions of $A$ range from 500 to 100000. Most of the matrices $A$ are dense and hence are not block-diagonal matrices, as discussed in Appendix C.1.

**Baselines.** We use the state-of-the-art exact algorithm Branch-and-Bound (Berk & Bertsimas, 2019) and the state-of-the-art polynomial-time approximation algorithm (in terms of multiplicative factor) Chan's algorithm (Chan et al., 2015) as our baselines.

**Notation.** We calculate the *approximation error* between two methods using the formula $(\mathrm{obj}_{\mathrm{Base}} - \mathrm{obj}_{\mathrm{Ours}})/\mathrm{obj}_{\mathrm{Base}} \times 100\%$, where $\mathrm{obj}_{\mathrm{Base}}$ is the objective value obtained by a baseline algorithm, while $\mathrm{obj}_{\mathrm{Ours}}$ is the objective value obtained by our framework integrated with the baseline algorithm. We calculate the *speedup factor* by dividing the runtime of a baseline algorithm by the runtime of Algorithm 5 integrated with the baseline algorithm.

**Results.** In Table 3, we report the performance of Algorithm 5 when integrated with Branch-and-Bound algorithm. We summarized the performance results for $k = 2, 5, 10, 15$ across various datasets. For each dataset, we report the average approximation error, average speedup factors, and

their corresponding standard deviations. Notably, the average speedup factor across all datasets is 100.50, demonstrating significant computational efficiency. Our theoretical analysis anticipates a trade-off between speedups and approximation errors; however, the empirical performance of our framework in terms of approximation accuracy is remarkably strong, with an average error of just 0.61% across all datasets.

In Table 4 we report the performance of Algorithm 5, in larger datasets with larger $k$'s, when integrated with Chan's algorithm (Chan et al., 2015). We summarized the performance results for $k = 200, 500, 1000, 2000$ across various datasets. Chan's algorithm terminates in time $\mathcal{O}(d^3)$ and returns an approximate solution to SPCA with a multiplicative factor $\min\{\sqrt{k}, d^{1/3}\}$. The average speedup factor is 6.00, and grows up to 14.51 as the dimension of the dataset grows, while the average error across all four datasets remains remarkably below 0%, meaning that our framework finds better solutions than the standalone algorithm.

These findings affirm the ability of our framework to handle diverse datasets and various problem sizes both efficiently and accurately. For a more detailed discussion, additional information on settings, and further empirical results, we refer readers to Appendix C.

Table 3: Summary of average approximation errors and average speedup factors for each dataset, compared to Branch-and-Bound. Standard deviations are reported in the parentheses.

| Dataset | Avg. Error (%) (Std. Dev) | Avg. Speedup (Std. Dev) |
|---|---|---|
| CovColonCov | 0.00 (0.00) | 25.84 (38.69) |
| LymphomaCov1 | 0.00 (0.00) | 22.64 (15.87) |
| Reddit1500 | 0.00 (0.00) | 365.12 (238.24) |
| Reddit2000 | 0.06 (0.13) | 278.69 (164.93) |
| LeukemiaCov | 0.00 (0.00) | 71.39 (83.17) |
| LymphomaCov2 | 4.82 (5.78) | 6.95 (4.35) |
| ProstateCov | 0.00 (0.00) | 19.90 (30.82) |
| ArceneCov | 0.00 (0.00) | 13.50 (14.17) |
| Overall | 0.61 (2.42) | 100.50 (163.60) |

Table 4: Summary of average approximation errors and average speedup factors for each dataset, compared to Chan's algorithm. Standard deviations are reported in the parentheses. Datasets are sorted in ascending order according to their dimensions.

| Dataset | Avg. Error (%) (Std. Dev) | Avg. Speedup (Std. Dev) |
|---|---|---|
| ArceneCov | -0.08 (0.10) | 0.86 (0.28) |
| GLI85Cov | -1.06 (1.22) | 1.58 (0.21) |
| GLABRA180Cov | -0.19 (0.26) | 7.07 (0.62) |
| DorotheaCov | -2.30 (0.62) | 14.51 (0.50) |
| Overall | -0.91 (1.11) | 6.00 (5.66) |

## 6 DISCUSSION

In this paper, we address the Sparse PCA problem by introducing an efficient framework that accommodates any off-the-shelf algorithm designed for Sparse PCA in a plug-and-play fashion. Our approach leverages matrix block-diagonalization, thereby facilitating significant speedups when integrated with existing algorithms. The implementation of our framework is very easy, enhancing its practical applicability. We have conducted a thorough theoretical analysis of both the approximation error and the time complexity associated with our method. Moreover, extensive empirical evaluations on large-scale datasets demonstrate the efficacy of our framework, which achieves considerable improvements in runtime with only minor trade-offs in approximation error.

Looking ahead, an important direction for future research involves extending our framework to handle the Sparse PCA problem with multiple principal components, while maintaining similar approximation and runtime guarantees. This extension could further broaden the applicability of our approach, addressing more complex scenarios in real-world applications.

## 7 ACKNOWLEDGEMENTS

A. Del Pia and D. Zhou are partially funded by AFOSR grant FA9550-23-1-0433. Any opinions, findings, and conclusions or recommendations expressed in this material are those of the authors and do not necessarily reflect the views of the Air Force Office of Scientific Research.

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

## A    ADDITIONAL RELATED WORK

In this section, we discuss additional related work. There exist many attempts in solving Sparse PCA in the literature, and we outlined only a selective overview of mainstream methods. Algorithms require worst-case exponential runtimes, including mixed-integer semidefinite programs (Bertsimas et al., 2022; Li & Xie, 2020; Cory-Wright & Pauphilet, 2022), mixed-integer quadratic programs (Dey et al., 2022b), non-convex programs (Zou et al., 2006; Dey et al., 2022a), and sub-exponential algorithms (Ding et al., 2023). On the other hand, there also exist a number of algorithms that take polynomial runtime to approximately solving Sparse PCA. To the best of our knowledge, the state-of-the-art approximation algorithm is proposed by Chan et al. (2015), in which an algorithm with multiplicative factor $\min\{\sqrt{k}, d^{-1/3}\}$ is proposed. There also exists tons of other (approximation) algorithms that have polynomial runtime, including semidefinite relaxations (Chowdhury et al., 2020), greedy algorithms (Li & Xie, 2020), low-rank approximations (Papailiopoulos et al., 2013), and fixed-rank Sparse PCA (Del Pia, 2022). Apart from the work we have mentioned, there also exists a streamline of work focusing on efficient estimation of sparse vectors in some specific statistical models. For instance, for a statistical model known as *Spiked Covariance model*, methods such as semidefinite relaxations (Amini & Wainwright, 2008; d'Orsi et al., 2020) and covariance thresholding (Deshp et al., 2016). Finally, there are also local approximation algorithms designed for Sparse PCA, and iterative algorithms are proposed (Yuan & Zhang, 2013; Hager et al., 2016).

The idea of decomposing a large-scale optimization problem into smaller sub-problems is a widely adopted approach in the optimization literature. For instance, Mazumder & Hastie (2012) leverages the connection between the thresholded input matrix and the support graph of the optimal solution in graphical lasso problem (GL), enabling significant computational speedups by solving sub-problems. Building on this, Fattahi & Sojoudi (2019) improves the computational efficiency of these subroutines by providing closed-form solutions under an acyclic assumption and analyzing the graph's edge structure. Wang et al. (2023) addresses GL in Gaussian graphical models through bridge-block decomposition and derives closed-form solutions for these blocks. In addition, several methods in the literature involve or are similar to matrix block-diagonalization. For instance, Feng et al. (2014) employ block-diagonalization for subspace clustering and segmentation problems by mapping data matrices into block-diagonal matrices through linear transformations. More recently, Han et al. (2023) introduce the sortedLSH algorithm to sparsify the softmax matrix, thus reducing the computational time of the attention matrix through random projection-based bucketing of "block indices". We remark that in all studies above, the underlying problems of interests, as well as the techniques, are different from ours, and the techniques developed therein cannot be directly applied to SPCA. A more related work (Devijver & Gallopin, 2018) explores block-diagonal covariance matrix estimation using hard thresholding techniques akin to ours. However, our framework diverges in two critical aspects: (i) They assume data is drawn from a Gaussian distribution, while we do not - we show in Section 4.2 that our method can by applied to *any* data inputs; and (ii) they use maximum likelihood estimation, whereas our Algorithm 5 involves iterative binary search to determine the optimal threshold $\varepsilon$.

## B    DEFERRED PROOFS

In this section, we provide proofs that we defer in the paper.

### B.1    PROOFS IN SECTION 3.2

**Lemma 1.** *Let $A \in \mathbb{R}^{d \times d}$ and $\varepsilon > 0$. Given input $(A, \varepsilon)$, Algorithm 1 outputs an $\varepsilon$-approximation of $A$, denoted as $\widetilde{A}$, such that $\mathrm{lbs}(\widetilde{A}) = \mathrm{int\,dim}(A, \varepsilon)$ in time $\mathcal{O}(d^2)$.*

*Proof of Lemma 1.* It is clear that Algorithm 1 has time complexity $\mathcal{O}(d^2)$, and it suffices to show that the output $\widetilde{A}$ of Algorithm 1 with input $(A, \varepsilon)$ satisfies $\mathrm{lbs}\left(\widetilde{A}\right) = \mathrm{int\,dim}(A, \varepsilon)$.

Suppose that $A^\star$ is an $\varepsilon$-approximation of $A$, and satisfies $\mathrm{lbs}(A^\star) = \mathrm{int\,dim}(A, \varepsilon)$. We intend to show $\mathrm{lbs}\left(\widetilde{A}\right) \leq \mathrm{lbs}(A^\star)$. WLOG we assume that $A^\star \in \mathcal{B}(A, \varepsilon)$ is a block-diagonal matrix with

blocks $A_1^\star, A_2^\star, \ldots, A_p^\star$, i.e., $A^\star = \text{diag}\left(A_1^\star, A_2^\star, \ldots, A_p^\star\right)$. Write $A = \begin{pmatrix} A_{11} & A_{12} & \ldots & A_{1p} \\ A_{21} & A_{22} & \ldots & A_{2p} \\ \vdots & \vdots & \vdots & \vdots \\ A_{p1} & A_{p2} & \ldots & A_{pp} \end{pmatrix}$

and $\widetilde{A} = \begin{pmatrix} \widetilde{A}_{11} & \widetilde{A}_{12} & \ldots & \widetilde{A}_{1p} \\ \widetilde{A}_{21} & \widetilde{A}_{22} & \ldots & \widetilde{A}_{2p} \\ \vdots & \vdots & \vdots & \vdots \\ \widetilde{A}_{p1} & \widetilde{A}_{p2} & \ldots & \widetilde{A}_{pp} \end{pmatrix}$, where $A_{ii}$, $\widetilde{A}_{ii}$, and $A_i^\star$ have the same number of rows and

columns, by the definition of $\mathcal{B}(A, \varepsilon)$, it is clear that $\|A_{ij}\|_\infty \leq \varepsilon$, for any $1 \leq i \neq j \leq p$. This further implies that for any $1 \leq i \neq j \leq p$, $\widetilde{A}_{ij}$ is a zero matrix by the procedure described in Algorithm 1, and therefore it is clear that $\text{lbs}\left(\widetilde{A}\right) \leq \text{lbs}\left(A^\star\right)$. By minimality of $A^\star$, we obtain that $\text{lbs}\left(\widetilde{A}\right) = \text{lbs}\left(A^\star\right)$. $\qquad\square$

### B.1.1 PROOFS IN SECTION 3.2.1

**Theorem 1.** *Let $A \in \mathbb{R}^{d \times d}$ be symmetric, $k \leq d$, and $\varepsilon > 0$. Denote by $A^\varepsilon$ the output of Algorithm 1 with input $(A, \varepsilon)$, $\widetilde{x}$ an optimal solution to SPCA with input $(A^\varepsilon, k)$, and $x^\star$ an optimal solution to SPCA with input $(A, k)$. Then, it follows that*

$$\left|(x^\star)^\top A x^\star - \widetilde{x}^\top A \widetilde{x}\right| \leq 2k \cdot \varepsilon.$$

*Proof of Theorem 1.* Note that

$$\left|(x^\star)^\top A x^\star - \widetilde{x}^\top A \widetilde{x}\right| \leq \left|(x^\star)^\top A x^\star - \widetilde{x}^\top A^\varepsilon \widetilde{x}\right| + \left|\widetilde{x}^\top (A^\varepsilon - A) \widetilde{x}\right|$$
$$\leq \left|(x^\star)^\top A x^\star - \widetilde{x}^\top A^\varepsilon \widetilde{x}\right| + k \cdot \|A - A^\varepsilon\|_\infty,$$

where the last inequality follows from Holder's inequality and the fact that $\|\widetilde{x}\|_1 \leq \sqrt{k} \cdot \|\widetilde{x}\|_2 = \sqrt{k}$. Therefore, it suffices to show that

$$\left|(x^\star)^\top A x^\star - \widetilde{x}^\top A^\varepsilon \widetilde{x}\right| \leq k \cdot \|A - A^\varepsilon\|_\infty.$$

Observe that

$$\left|(x^\star)^\top A x^\star - \widetilde{x}^\top A^\varepsilon \widetilde{x}\right| = \left| \max_{\substack{\|x\|_2=1, \\ \|x\|_0 \leq k}} x^\top A x - \max_{\substack{\|x\|_2=1, \\ \|x\|_0 \leq k}} x^\top A^\varepsilon x \right|$$
$$\leq \max_{\substack{\|x\|_2=1, \\ \|x\|_0 \leq k}} \left|x^\top (A^\varepsilon - A) x\right|$$
$$\leq k \cdot \|A - A^\varepsilon\|_\infty,$$

where the first inequality follows from the following facts:

$$\max_{\substack{\|x\|_2=1, \\ \|x\|_0 \leq k}} x^\top A x - \max_{\substack{\|x\|_2=1, \\ \|x\|_0 \leq k}} x^\top A^\varepsilon x \leq \max_{\substack{\|x\|_2=1, \\ \|x\|_0 \leq k}} x^\top (A - A^\varepsilon) x$$

$$\max_{\substack{\|x\|_2=1, \\ \|x\|_0 \leq k}} x^\top A^\varepsilon x - \max_{\substack{\|x\|_2=1, \\ \|x\|_0 \leq k}} x^\top A x \leq \max_{\substack{\|x\|_2=1, \\ \|x\|_0 \leq k}} x^\top (A^\varepsilon - A) x$$

Finally, combining the fact that $\|A - A^\varepsilon\|_\infty \leq \varepsilon$, we are done. $\qquad\square$

**Theorem 2.** *Let $\widetilde{A} = \text{diag}(\widetilde{A}_1, \widetilde{A}_2, \ldots, \widetilde{A}_p)$ be a symmetric block-diagonal matrix. Denote $\mathsf{OPT}$ to be the optimal value to SPCA with input pair $(\widetilde{A}, k)$. Let $\mathsf{OPT}_i$ to be the optimal value to SPCA with input pair $(\widetilde{A}_i, k)$, for $i \in [p]$. Then, one has $\mathsf{OPT} = \max_{i \in [p]} \mathsf{OPT}_i$.*

*Proof of Theorem 2.* It is clear that $\mathsf{OPT} \geq \max_{i \in [p]} \mathsf{OPT}_i$. It remains to show the reverse, i.e., $\mathsf{OPT} \leq \max_{i \in [p]} \mathsf{OPT}_i$.

Suppose not, we have $\mathsf{OPT} > \max_{i \in [p]} \mathsf{OPT}_i$. Then, there must exist principal submatrices $\widetilde{A}'_1, \widetilde{A}'_2, \ldots, \widetilde{A}'_p$ contained in $\widetilde{A}_1, \widetilde{A}_2, \ldots, \widetilde{A}_p$ such that $\mathsf{diag}(\widetilde{A}'_1, \widetilde{A}'_2, \ldots, \widetilde{A}'_p)$ is a $k \times k$ matrix, and

$$\mathsf{OPT} = \max_{\|x\|_2=1} x^\top \mathsf{diag}(\widetilde{A}'_1, \widetilde{A}'_2, \ldots, \widetilde{A}'_p)x > \max_{i \in [p]} \mathsf{OPT}_i.$$

This gives a contradiction, as

$$\max_{i \in [p]} \mathsf{OPT}_i = \max_{i \in [p]} \max_{\substack{\|x\|_2=1, \\ \|x\|_0 \leq k}} x^\top \widetilde{A}_i x \geq \max_{\|x\|_2=1} x^\top \mathsf{diag}(\widetilde{A}'_1, \widetilde{A}'_2, \ldots, \widetilde{A}'_p)x,$$

where the inequality follows from the fact that

$$\max_{\|x\|_2=1} x^\top \mathsf{diag}(\widetilde{A}'_1, \widetilde{A}'_2, \ldots, \widetilde{A}'_p)x = \max_{i \in [p]} \max_{\|x\|_2=1} x^\top \widetilde{A}'_i x = \max_{i \in [p]} \max_{\substack{\|x\|_2=1 \\ \|x\|_0 \leq k}} x^\top \widetilde{A}_i x = \max_{i \in [p]} \mathsf{OPT}_i.$$

$\square$

**Theorem 3.** *Let $A \in \mathbb{R}^{d \times d}$ be symmetric and positive semidefinite, let $k$ be a positive integer, and denote by $x^\star \in \mathbb{R}^d$ an optimal solution to SPCA with input $(A, k)$. Suppose that an algorithm $\mathcal{A}$ for SPCA with input $(A, k)$ finds an approximate solution $x \in \mathbb{R}^d$ to SPCA with multiplicative factor $m(k, d) \geq 1$ and additive error $a(k, d) \geq 0$, i.e., one has $x^\top A x \geq (x^\star)^\top A x^\star / m(k, d) - a(k, d)$. Furthermore, we assume that $m(k, d)$ and $a(k, d)$ is non-decreasing with respect to $d$. For $\varepsilon > 0$, denote by $y \in \mathbb{R}^d$ be the output of Algorithm 3 with input tuple $(A, k, \varepsilon, \mathcal{A})$. Then, one has*

$$y^\top A y \geq \frac{(x^\star)^\top A x^\star}{m(k, \mathsf{int\,dim}(A, \varepsilon))} - a(k, \mathsf{int\,dim}(A, \varepsilon)) - \frac{1}{m(k, \mathsf{int\,dim}(A, \varepsilon))} \cdot k\varepsilon.$$

*Proof of Theorem 3.* In the proof, we use the same notation as in Algorithm 3. We assume WLOG that the thresholded matrix $A^\varepsilon$ is a block-diagonal matrix, i.e., $A^\varepsilon = \mathsf{diag}(A_1^\varepsilon, A_2^\varepsilon, \ldots, A_p^\varepsilon)$, where $A_i^\varepsilon := A_{S_i, S_i}^\varepsilon$. We also define $\widetilde{A} := \mathsf{diag}\left(\widetilde{A}_1, \widetilde{A}_2, \ldots, \widetilde{A}_p\right)$, with $\widetilde{A}_i := A_{S_i, S_i}$. It is clear that $\left\|A - \widetilde{A}\right\|_\infty \leq \varepsilon$. Denote by $\widetilde{x}^\star \in \mathbb{R}^d$ an optimal solution to SPCA with input $(\widetilde{A}, k)$. By Theorem 2, one can assume WLOG that $\widetilde{x}^\star$ has zero entries for indices greater than $d_1 := |S_1|$. In other words, $\widetilde{x}^\star$ is "found" in the block $A_1$. Recall that we denote $x_i$ the solution found by $\mathcal{A}$ with input $\left(\widetilde{A}_i, k\right)$. Then, it is clear that

$$\max_{i \in [p]} x_i^\top \widetilde{A}_i x_i \geq x_1^\top \widetilde{A}_1 x_1$$

$$\geq \frac{1}{m(k, d_1)} \cdot (\widetilde{x}^\star)^\top \widetilde{A} \widetilde{x}^\star - a(k, d_1)$$

$$\geq \frac{1}{m(k, \mathsf{int\,dim}(A, \varepsilon))} \cdot (\widetilde{x}^\star)^\top \widetilde{A} \widetilde{x}^\star - a(k, \mathsf{int\,dim}(A, \varepsilon)).$$

Recall that $y_i$ is the reconstructed solution. By the proof of Theorem 1, one obtains that

$$
\begin{aligned}
&\max_{i \in [p]} y_i^\top A y_i \\
&= \max_{i \in [p]} x_i^\top \widetilde{A}_i x_i \\
&\geq \frac{1}{m(k, \mathsf{int\,dim}(A, \varepsilon))} \cdot (\widetilde{x}^\star)^\top \widetilde{A} \widetilde{x}^\star - a(k, \mathsf{int\,dim}(A, \varepsilon)) \\
&\geq \frac{1}{m(k, \mathsf{int\,dim}(A, \varepsilon))} \cdot \left((x^\star)^\top A x^\star - k \left\|A - \widetilde{A}\right\|_\infty\right) - a(k, \mathsf{int\,dim}(A, \varepsilon)).
\end{aligned}
\tag{1}
$$

$\square$

### B.1.2 Proofs in Section 3.2.2

**Proposition 1.** *Let $A \in \mathbb{R}^{d \times d}$, and let $k$ be a positive integer. Suppose that for a specific algorithm $\mathcal{A}$, the time complexity of using $\mathcal{A}$ for (approximately) solving SPCA with input $(A, k)$ is a function $g(k, d)$ which is convex and non-decreasing with respect to $d$. We assume that $g(k, 1) = 1$. Then, for a given threshold $\varepsilon > 0$, the runtime of Algorithm 3 with input tuples $(A, k, \varepsilon, \mathcal{A})$ is at most*

$$\mathcal{O}\left(\left\lceil \frac{d}{\mathrm{int}\,\dim(A, \varepsilon)} \right\rceil \cdot g\left(k, \mathrm{int}\,\dim(A, \varepsilon)\right) + d^2\right)$$

*Proof of Proposition 1.* Suppose that the largest block size of $\widetilde{A} \in \mathcal{B}(A, \varepsilon)$ is equal to $\mathrm{int}\,\dim(A, \varepsilon)$, and $\widetilde{A}$ can be sorted to a block-diagonal matrix with $p$ blocks $\widehat{A}_1, \widehat{A}_2, \ldots, \widehat{A}_p$ and block sizes $d_1, d_2, \ldots, d_p$, respectively. WLOG we assume that $p < d$, otherwise the statement holds trivially. It is clear that the time complexity of running line 5 - 8 in Algorithm 3 is upper bounded by a constant multiple of

$$\max_{\substack{1 \leq d_i \leq \mathrm{int}\,\dim(A, \varepsilon) \\ \sum_{i=1}^{m} d_i = d}} \sum_{i=1}^{p} g(k, d_k). \tag{2}$$

Since $k$ is fixed, the optimization problem (2) is maximizing a convex objective function over a polytope

$$P := \left\{ (d_1, d_2, \ldots, d_p) \in \mathbb{R}^p : 1 \leq d_i \leq \mathrm{int}\,\dim(A, \varepsilon), \ \forall i \in [p], \ \sum_{i=1}^{p} d_i = d \right\}.$$

Therefore, the optimum to (2) is obtained at an extreme point of $P$. It is clear that an extreme point $(d_1^\star, d_2^\star, \ldots, d_p^\star)$ of $P$ is active at a linearly independent system, i.e., among all $d_i^\star$'s, there are $p - 1$ of them must be equal to 1 or $\mathrm{int}\,\dim(A, \varepsilon)$, and the last one is taken such that the euqality $\sum_{i=1}^{p} d_i^\star = d$ holds. Since there could be at most $\lfloor d / \mathrm{int}\,\dim(A, \varepsilon) \rfloor$ many blocks in $\widetilde{A}$ with block size equal to $\mathrm{int}\,\dim(A, \varepsilon)$, and that $g(k, d)$ is increasing with respect to $d$, we obtain that the optimum of the optimization problem (2) is upper bounded by

$$\left\lceil \frac{d}{\mathrm{int}\,\dim(A, \varepsilon)} \right\rceil \cdot g\left(k, \mathrm{int}\,\dim(A, \varepsilon)\right) + \left( p - \left\lfloor \frac{d}{\mathrm{int}\,\dim(A, \varepsilon)} \right\rfloor \right)$$

$$< \left\lceil \frac{d}{\mathrm{int}\,\dim(A, \varepsilon)} \right\rceil \cdot g\left(k, \mathrm{int}\,\dim(A, \varepsilon)\right) + d.$$

We are then done by noticing the fact that lines 3 - 4 in Algorithm 3 have a runtime bounded by $\mathcal{O}(d^2)$. $\qquad\square$

### B.2 Proofs in Section 4.1

To establish a connection with robust statistics, we begin by expressing each element of the input matrix $A$ in Model 1 as follows:

$$A_{ij} = \widetilde{A}ij + Eij := \widetilde{A}ij + \sigma^2 Zij, \quad i \leq j,$$

where $Z_{ij}$ is an i.i.d. centered $(\varrho^2/\sigma^2)$-sub-Gaussian random variable with unit variance. Considering that $\widetilde{A}$ is block-diagonal and, according to (ii) in Model 1, has at most $\lceil d/d^\star \rceil \cdot (d^\star)^2 = \mathcal{O}(d^{1+\alpha})$ nonzero entries, the methodology proposed by Comminges et al. (2021) can be employed to estimate $\sigma^2$ accurately using Algorithm 4.

Inspired by the methods proposed in Comminges et al. (2021), we show that the number $\bar{\sigma}$ found in Algorithm 4 could provide an constant approximation ratio to the true variance $\sigma^2$ in Model 1 with high probability:

**Proposition 3** (adapted from Proposition 1 in Deshp et al. (2016))**.** *Consider Model 1, and denote $\mathcal{P}_Z$ to be the distribution of the random variable $E_{ij}/\sigma^2$. Let $\bar{\sigma}^2$ be the same number found in*

*Algorithm 4 with input tuple $(A, C, \alpha)$. There exist an absolute constant $c > 0$ such that for the dimension $d$ of the input matrix $A$ satisfying*

$$\log d > \log \left[ (4C + 4) \cdot \left( \frac{\log (8C + 8)}{\frac{c\sigma^2}{2\varrho^2} \cdot \left( \frac{\sigma^2}{2\varrho^2} + 1 \right)} + 1 \right) \right] / (1 - \alpha)$$

*we have that for distribution $\mathcal{P}_Z$ of $Z_{ij}$, the variance $\sigma^2$, one has a uniform lower bound on the accuracy of estimation:*

$$\inf_{\mathcal{P}_Z} \inf_{\sigma > 0} \inf_{\|\widetilde{A}\|_0 \leq Cd^{1+\alpha}} \mathbb{P}_{\mathcal{P}_Z, \sigma, \widetilde{A}} \left( \frac{1}{2} \leq \frac{\bar{\sigma}^2}{\sigma^2} \leq \frac{3}{2} \right) \geq 1 - \exp \left\{ -\frac{d^{\alpha+1}}{4C + 4} \right\}.$$

*Proof of Proposition 3.* Recall that we write

$$A_{ij} = \widetilde{A}ij + Eij := \widetilde{A}ij + \sigma^2 Zij, \ i \leq j,$$

where $Z_{ij}$ is an i.i.d. centered $(\varrho^2/\sigma^2)$-sub-Gaussian random variable with unit variance. Since $d^\star \leq Cd^\alpha$ for some $\alpha \in (0, 1)$, WLOG we assume that $d^\star = Cd^\alpha$ for some $C > 0$, and for simplicity we assume that $d^{1+\alpha}$ and $C$ are all integers to avoid countless discussions of integrality. By Algorithm 4, one divides the index set $\{(i, j) \in [d] \times [d] : i \leq j\}$ into $m = (2C + 2)d^{1+\alpha}$ disjoint subsets $B_1, B_2, \ldots, B_m$, and each $B_i$ has cardinality at lease $k := \lfloor (d^2 + d)/(2m) \rfloor$. Denote the set $J$ to be the set of indices $i$ such that the set $B_i$ contains only indices such that for any $(k, l) \in B_i$, $\widetilde{A}_{kl} = 0$. It is clear that $|J|$ is lower bounded by $m - Cd^{\alpha+1} = (2C + 1)d^{1+\alpha}$.

By Bernstein's inequality (see, e.g., Corollary 2.8.3 in Vershynin (2018)), it is clear that there exists an absolute constant $c > 0$ such that

$$\mathbb{P} \left( \left| \frac{1}{|B_i|} \sum_{(k,l) \in B_i} E_{kl}^2 - \sigma^2 \right| > \frac{\sigma^2}{2} \right) = \mathbb{P} \left( \left| \frac{1}{|B_i|} \sum_{(k,l) \in B_i} Z_{kl}^2 - 1 \right| > \frac{1}{2} \right)$$

$$\leq 2 \exp \left\{ -c \min \left( \frac{\sigma^4}{4\varrho^4}, \frac{\sigma^2}{2\varrho^2} \right) |B_i| \right\}$$

$$\leq 2 \exp \left\{ -c \min \left( \frac{\sigma^4}{4\varrho^4}, \frac{\sigma^2}{2\varrho^2} \right) \left\lfloor \frac{d^2 + d}{2m} \right\rfloor \right\}.$$

Then, we denote $I := [\sigma^2/2, 3\sigma^2/2]$, denote $A_i$ to be the random event $\left\{ \frac{1}{|B_i|} \sum_{(k,l) \in B_i} A_{kl}^2 \notin I \right\}$, and denote $\mathbb{1}_{A_i}$ to be the indicator function of $A_i$. One obtains that

$$\mathbb{P} \left( \bar{\sigma}^2 \notin I \right) \leq \mathbb{P} \left( \sum_{i=1}^m \mathbb{1}_{A_i} \geq \frac{m}{2} \right)$$

$$= \mathbb{P} \left( \sum_{i \in J} \mathbb{1}_{A_i} \geq \frac{m}{2} - \sum_{i \notin J} \mathbb{1}_{A_i} \right)$$

$$\leq \mathbb{P} \left( \sum_{i \in J} \mathbb{1}_{A_i} \geq \frac{m}{2} - Cd^{\alpha+1} \right)$$

Denote

$$t := \frac{m}{2} - Cd^{\alpha+1} - 2|J| \exp \left\{ -c \min \left( \frac{\sigma^4}{4\varrho^4}, \frac{\sigma^2}{2\varrho^2} \right) \left\lfloor \frac{d^2 + d}{2m} \right\rfloor \right\},$$

and by Hoeffding's inequality, one obtains that

$$\mathbb{P}\left(\sum_{i \in J} \mathbb{1}_{A_i} \geq \frac{m}{2} - Cd^{\alpha+1}\right)$$

$$\leq \mathbb{P}\left(\sum_{i \in J}(\mathbb{1}_{A_i} - \mathbb{E}\mathbb{1}_{A_i}) \geq \frac{m}{2} - Cd^{\alpha+1} - 2|J|\exp\left\{-c\min\left(\frac{\sigma^4}{4\varrho^4}, \frac{\sigma^2}{2\varrho^2}\right)\left\lfloor\frac{d^2+d}{2m}\right\rfloor\right\}\right)$$

$$= \mathbb{P}\left(\sum_{i \in J}(\mathbb{1}_{A_i} - \mathbb{E}\mathbb{1}_{A_i}) \geq t\right)$$

$$\leq \exp\left\{-\frac{2t^2}{|J|}\right\}$$

Since $m = (2C+2)d^{\alpha+1}$, it is clear that $m \geq |J| \geq (2C+1)d^{\alpha+1}$, and

$$t \geq d^{\alpha+1}\left(1 - 2(2C+2)\exp\left\{-c\min\left(\frac{\sigma^4}{4\varrho^4}, \frac{\sigma^2}{2\varrho^2}\right)\lfloor d^{1-\alpha}/(4C+4)\rfloor\right\}\right) \geq d^{\alpha+1} \cdot \frac{1}{2},$$

where the last inequality follows from the fact that

$$\log d > \log\left[(4C+4) \cdot \frac{\log(8C+8)}{\frac{c\sigma^2}{2\varrho^2} \cdot \left(\frac{\sigma^2}{2\varrho^2}+1\right)}\right] / (1-\alpha).$$

Hence, the probability of $\bar{\sigma}^2 \notin I$ is upper bounded by $\exp\{-d^{\alpha+1}/(4C+4)\}$. $\qquad\square$

**Proposition 2.** *Consider Model 1, and denote by $\bar{\varepsilon}$ the output of Algorithm 4 with input tuple $(A, C, \alpha, u)$. Let $k \leq d$ be a positive integer, and assume that $d$ satisfies that $d^{1-\alpha} > C_0 \cdot (C+1) \cdot u\log(8C+8)$ for some large enough absolute constant $C_0 > 0$. Then, the following holds with probability at least $1 - 2d^{-1} - \exp\{2\log d - d^{1+\alpha}/(4C+4)\}$:*

*(i) Denote by $\widetilde{x}^\star$ the optimal solution to SPCA with input $(\widetilde{A}, k)$. For an (approximation) algorithm $\mathcal{A}$ with multiplicative factor $m(k,d) \geq 1$ and additive error $a(k,d) \geq 0$, where the functions $m$ and $a$ is non-deceasing with respect to $d$, the output $y$ of Algorithm 3 with input tuple $(A, k, \bar{\varepsilon}, \mathcal{A})$ satisfies that*

$$y^\top \widetilde{A} y \geq \frac{(\widetilde{x}^\star)^\top \widetilde{A} \widetilde{x}^\star}{m(k, d^\star)} - a(k, d^\star) - \left(1 + \frac{2}{m(k, d^\star)}\right) \cdot k\bar{\varepsilon}.$$

*(ii) If, in addition, the time complexity of using $\mathcal{A}$ for (approximately) solving SPCA with input $(A, k)$ is a function $g(k, d)$ which is convex and non-decreasing with respect to $d$, and satisfies $g(k, 1) = 1$. Then, the runtime of Algorithm 3 with input tuples $(A, k, \varepsilon, \mathcal{A})$ is at most*

$$\mathcal{O}\left(\left\lceil\frac{d}{d^\star}\right\rceil \cdot g(k, d^\star) + d^2\right)$$

*Proof of Proposition 2.* We first show that, the probability that $\|E\|_\infty \leq \bar{\varepsilon}$ is at least $1 - 2d^{-1} - \exp\left\{2\log d - \frac{d^{1+\alpha}}{4C+4}\right\}$. This comes directly from Proposition 3 that $\bar{\sigma}^2$ satisfies that $\frac{1}{2} \leq \frac{\bar{\sigma}^2}{\sigma^2} \leq \frac{3}{2}$ with probability at least $1 - \exp\{-d^{1+\alpha}/(4C+4)\}$, the definition of $E_{kl}$ being i.i.d. $\varrho$-sub-Gaussian variables, and a union bound argument. Indeed, define the random event $A := \{\frac{1}{2} \leq \frac{\bar{\sigma}^2}{\sigma^2} \leq \frac{3}{2}\}$, one

can see that

$$
\begin{aligned}
\mathbb{P}\left(|E_{kl}| > \bar{\varepsilon}\right) &= \mathbb{P}\left(|E_{kl}| > \bar{\varepsilon} \mid A\right)\mathbb{P}(A) + \mathbb{P}\left(|E_{kl}| > \bar{\varepsilon} \mid A^c\right)\mathbb{P}(A^c) \\
&= \mathbb{P}\left(|E_{kl}| > 2\sqrt{3}u\bar{\sigma}\sqrt{\log d} \mid A\right)\mathbb{P}(A) + \mathbb{P}\left(|E_{kl}| > \bar{\varepsilon} \mid A^c\right)\mathbb{P}(A^c) \\
&\le \mathbb{P}\left(|E_{kl}| > \sqrt{6}u\sigma\sqrt{\log d} \mid A\right)\mathbb{P}(A) + \mathbb{P}(A^c) \\
&\le \mathbb{P}\left(|E_{kl}| > \sqrt{6}u\sigma\sqrt{\log d}\right) + \exp\left\{-\frac{d^{1+\alpha}}{4C+4}\right\} \\
&\le 2\exp\left\{-\frac{6u^2\sigma^2}{2\varrho^2}\log d\right\} + \exp\left\{-\frac{d^{1+\alpha}}{4C+4}\right\} \\
&= 2d^{-3} + \exp\left\{-\frac{d^{1+\alpha}}{4C+4}\right\}.
\end{aligned}
$$

By union bound, one has

$$
\mathbb{P}\left(\|E\|_\infty > \bar{\varepsilon}\right) \le d^2 \cdot \mathbb{P}\left(|E_{kl}| > \bar{\varepsilon}\right) \le 2d^{-1} + \exp\left\{2\log d - \frac{d^{1+\alpha}}{4C+4}\right\}.
$$

This implies that, with high probability, $\operatorname{int}\dim(A, \bar{\varepsilon}) \le d^\star$. In the remainder of the proof, we will assume that the random event $\|E\|_\infty \le \bar{\varepsilon}$ holds true.

We are now ready to show the desired approximation error. Denote by $x^\star$ the optimal solution to SPCA with input $(A, k)$, by the proof of Theorem 1, it is clear that

$$
(x^\star)^\top A x^\star \ge (\widetilde{x}^\star)^\top \widetilde{A}\widetilde{x}^\star - k\bar{\varepsilon},
$$

due to the fact that $\left\|\widetilde{A} - A\right\|_\infty \le \bar{\varepsilon}$. Combining with Theorem 3, we have

$$
\begin{aligned}
y^\top \widetilde{A}y &\ge y^\top A y - k \cdot \bar{\varepsilon} \\
&\ge \frac{(x^\star)^\top A x^\star}{m\left(k, \operatorname{int}\dim(A, \varepsilon)\right)} - a\left(k, \operatorname{int}\dim(A, \varepsilon)\right) - \left(1 + \frac{1}{m\left(k, \operatorname{int}\dim(A, \varepsilon)\right)}\right) \cdot k\bar{\varepsilon} \\
&\ge \frac{(\widetilde{x}^\star)^\top \widetilde{A}\widetilde{x}^\star}{m\left(k, \operatorname{int}\dim(A, \varepsilon)\right)} - a\left(k, \operatorname{int}\dim(A, \varepsilon)\right) - \left(1 + \frac{2}{m\left(k, \operatorname{int}\dim(A, \varepsilon)\right)}\right) \cdot k\bar{\varepsilon}.
\end{aligned}
$$

Finally, we obtain the desired runtime complexity via Proposition 1. $\qquad\square$

### B.3 Proofs in Section 4.2

**Theorem 4.** *Let $A \in \mathbb{R}^{d \times d}$ be symmetric and positive semidefinite, and let $k$ be a positive integer. Suppose that an algorithm $\mathcal{A}$ for SPCA with input $(A, k)$ finds an approximate solution $x \in \mathbb{R}^d$ to SPCA such that has multiplicative factor $m(k, d) \ge 1$ and additive error $a(k, d) \ge 0$, with $m(k, d)$ and $a(k, d)$ being non-decreasing with respect to $d$. Suppose that Algorithm 5 with input $(A, k, \delta, \mathcal{A}, d_0)$ terminates with $\mathsf{OPT}(\varepsilon^\star) \ge 0$. Then one has*

$$
\mathsf{OPT}(\varepsilon^\star) \ge \frac{1}{m(k, d_0)} \cdot \mathsf{OPT}(0) - a(k, d_0) - \frac{1}{m(k, d_0)} \cdot k\varepsilon^\star.
$$

*Suppose that for $\mathcal{A}$, the time complexity of using $\mathcal{A}$ for (approximately) solving SPCA with input $(A, k)$ is a function $g(k, d)$ which is convex and non-decreasing with respect to $d$, $g(k, 1) = 1$, and $g(k, 0) = 0$. The runtime for Algorithm 5 is at most*

$$
\mathcal{O}\left(\log\left(\frac{\|A\|_\infty}{\delta}\right) \cdot \left(\left\lceil\frac{d}{d_0}\right\rceil \cdot g(k, d_0) + d^2\right)\right)
$$

*Proof of Theorem 4.* We will use Theorem 3 and Proposition 1 to prove this theorem.

We first prove the approximation bound. First note that $\left\| A - A^{\varepsilon^\star} \right\|_\infty \le \varepsilon^\star$. By Theorem 3,

$$\mathsf{OPT}(\varepsilon^\star) \ge \frac{(x^\star)^\top A x^\star}{m\left(k, \operatorname{int} \dim(A, \varepsilon^\star)\right)} - a\left(k, \operatorname{int} \dim(A, \varepsilon^\star)\right) - \frac{1}{m\left(k, \operatorname{int} \dim(A, \varepsilon^\star)\right)} \cdot k\varepsilon^\star$$

$$= \frac{\mathsf{OPT}(0)}{m\left(k, \operatorname{int} \dim(A, \varepsilon^\star)\right)} - a\left(k, \operatorname{int} \dim(A, \varepsilon^\star)\right) - \frac{1}{m\left(k, \operatorname{int} \dim(A, \varepsilon^\star)\right)} \cdot k\varepsilon^\star.$$

Note that in Algorithm 5, the algorithm is guaranteed to terminate with an $\varepsilon^\star$ such that $\operatorname{int} \dim(A, \varepsilon^\star) \le d_0$. Since the functions $m(k, d)$ and $a(k, d)$ are all non-decreasing with respect to $d$, we have that $m\left(k, \operatorname{int} \dim(A, \varepsilon^\star)\right) \le m(k, d_0)$ and $a\left(k, \operatorname{int} \dim(A, \varepsilon^\star)\right) \le a(k, d_0)$, and therefore we obtain our desired approximation bound

$$\mathsf{OPT}(\varepsilon^\star) \ge \frac{\mathsf{OPT}(0)}{m(k, d_0)} - a(k, d_0) - \frac{1}{m(k, d_0)} \cdot k\varepsilon^\star.$$

Here, the inequality holds due to the fact that (i) if $\mathsf{OPT}(0) \le k\varepsilon^\star$, then since $\mathsf{OPT}(\varepsilon^\star)$ is non-negative, the inequality trivially holds; (ii) if $\mathsf{OPT}(0) > k\varepsilon^\star$, then the inequality also holds since $a(k, d)$ and $m(k, d)$ are non-decreasing with respect to $d$.

Then, we show the time complexity. By Proposition 1, it is clear that the runtime for a single iteration in the while loop in Algorithm 5 is upper bounded by

$$\mathcal{O}\left(\left\lceil \frac{d}{\operatorname{int} \dim(A, \varepsilon)} \right\rceil \cdot g\left(k, \operatorname{int} \dim(A, \varepsilon)\right) + d^2\right), \tag{3}$$

for some $\varepsilon \ge 0$ such that $\operatorname{int} \dim(A, \varepsilon) \le d_0$. Let $\varepsilon_1 \ge \varepsilon_2$ be two positive numbers, and it is clear that $\operatorname{int} \dim(A, \varepsilon_1) \le \operatorname{int} \dim(A, \varepsilon_2)$. Since

$$\operatorname{int} \dim(A, \varepsilon_1) = \frac{\operatorname{int} \dim(A, \varepsilon_1)}{\operatorname{int} \dim(A, \varepsilon_2)} \cdot \operatorname{int} \dim(A, \varepsilon_2) + \left(1 - \frac{\operatorname{int} \dim(A, \varepsilon_1)}{\operatorname{int} \dim(A, \varepsilon_2)}\right) \cdot 0,$$

and by convexity of $g(k, d)$, along with the fact that $g(k, 0) = 0$, we obtain that

$$g(k, \operatorname{int} \dim(A, \varepsilon_1)) \le \frac{\operatorname{int} \dim(A, \varepsilon_1)}{\operatorname{int} \dim(A, \varepsilon_2)} \cdot g(k, \operatorname{int} \dim(A, \varepsilon_2)) + 0.$$

This implies that for any $\varepsilon \ge 0$ such that $\operatorname{int} \dim(A, \varepsilon) \le d_0$, we have that

$$\frac{d}{\operatorname{int} \dim(A, \varepsilon_1)} \cdot g(k, \operatorname{int} \dim(A, \varepsilon_1)) \le \frac{d}{d_0} \cdot g(k, d_0),$$

and thus

$$\left\lceil \frac{d}{\operatorname{int} \dim(A, \varepsilon)} \right\rceil \cdot g(k, \operatorname{int} \dim(A, \varepsilon))$$

$$\le \frac{d}{\operatorname{int} \dim(A, \varepsilon)} \cdot g(k, \operatorname{int} \dim(A, \varepsilon)) + g(k, \operatorname{int} \dim(A, \varepsilon))$$

$$\le \frac{d}{d_0} \cdot g(k, d_0) + g(k, \operatorname{int} \dim(A, \varepsilon))$$

$$\le \frac{d}{d_0} \cdot g(k, d_0) + g(k, d_0) \le 2 \left\lceil \frac{d}{d_0} \right\rceil \cdot g(k, d_0).$$

Combining with (3), and the fact that at most $\mathcal{O}(\log(\|A\|_\infty / \delta))$ iterations would be executed in the while loop in Algorithm 5, we are done. $\qquad\square$

## C ADDITIONAL EMPIRICAL SETTINGS AND RESULTS

In this section, we report additional empirical settings in Appendix C.1 and detailed empirical results in Appendices C.2 and C.3. We report empirical results under Model 1 in Appendix C.4. In Appendix C.5, we discuss impacts of parameters $d_0$ and $\delta$ in Algorithm 5.

## C.1 EMPIRICAL SETTINGS

**Datasets.** We summarize the dimensions and percentage of non-zero entries in the input matrices in Table 5. It is important to note that more than 70% of these input covariance matrices are dense. We utilize these matrices to evaluate the practicality of our approach, demonstrating its effectiveness irrespective of whether the input matrix has a block-diagonal structure.

Table 5: Dimensions for each dataset and percentage of non-zero entries.

| Dataset | Dimension ($d$) | Percentage of non-zero entries |
|---|---|---|
| CovColonCov | 500 | 100% |
| LymphomaCov1 | 500 | 100% |
| Reddit1500 | 1500 | 5.30% |
| Reddit2000 | 2000 | 6.20% |
| LeukemiaCov | 3571 | 100% |
| LymphomaCov2 | 4026 | 100% |
| ProstateCov | 6033 | 100% |
| ArceneCov | 10000 | 100% |
| GLI85Cov | 22283 | 100% |
| GLABRA180Cov | 49151 | 100% |
| DorotheaCov | 100000 | 77.65% |

**Parameters.** (i) When integrated Algorithm 5 with Branch-and-Bound algorithm: We choose the parameter $\bar{\varepsilon} := 1$, $a := 0$, $b := \|A\|_\infty$, $d_0 := 30$, and $\delta := 0.01 \cdot \|A\|_\infty$. (ii) When integrated Algorithm 5 with Chan's algorithm: We choose the parameter $\bar{\varepsilon} := 1$, $a := 0$, $b := \|A\|_\infty$, $d_0 := 2k$ (twice the sparsity constant), and $\delta := 0.01 \cdot \|A\|_\infty$.

**Early Stopping.** For experiments conducted with a positive semidefinite input matrix $A$, we add a very simple step in Algorithm 5 - we set an extra stopping criteria on Algorithm 5, which breaks the while loop as soon as a problem with largest block size $d_0$ has been solved. This additional step helps further reduce computational time of Algorithm 5.

**Compute resources.** We conducted all tests on a computing cluster equipped with 36 Cores (2x 3.1G Xeon Gold 6254 CPUs) and 768 GB of memory.

## C.2 EMPIRICAL RESULTS WHEN INTEGRATED WITH BRANCH-AND-BOUND ALGORITHM

In this section, we provide the detailed empirical results of our framework when integrated with Branch-and-Bound algorithm (previously summarized in Table 3), in Table 6. We provide statistics related to the best solution in Table 7, including the ratio of corresponding $\varepsilon^\star$ (the best threshold found) and $\|A\|_\infty$, the percentage of zeros in the matrix $A^{\varepsilon^\star}$, and the *Jaccard index* between the solution obtained by baseline algorithm and that obtained by Algorithm 5, which is defined as

$$\text{Jaccard index} := \frac{|\operatorname{supp}(x_{\text{Base}}) \cap \operatorname{supp}(x_{\text{Ours}})|}{|\operatorname{supp}(x_{\text{Base}}) \cup \operatorname{supp}(x_{\text{Ours}})|}.$$

Here, $x_{\text{Base}}$ denotes the solution obtained by the baseline algorithm, which refers to Branch-and-Bound algorithm in this section and Chan's algorithm in Appendix C.3, and $x_{\text{Ours}}$ denotes the solution obtained by Algorithm 5 when integrated with the baseline algorithm. We also summarize the average approximation errors and average speedup factors for with respect to $k$ in Table 8.

From the data presented in Table 6, it is evident that our framework achieves exceptional performance metrics. The median approximation error is 0, with the 90th percentile approximation error not exceeding 0.19%. Additionally, the median speedup factor is recorded at 20.87. These statistics further underscore the efficiency and effectiveness of our framework, demonstrating its capability to provide substantial computational speedups while maintaining small approximation errors.

From the data in Table 7, we conclude that the Jaccard index is oftentimes one, and the change of Jaccard index aligns with the change of the approximation error. Additionally, for each dataset, the relative threshold value $\varepsilon^\star / \|A\|_\infty$ and the percentage of zeros in $A^{\varepsilon^\star}$ remain relatively stable across

Table 6: Comparison of runtime and objective values on real-world datasets, between Branch-and-Bound and Algorithm 5 integrated with Branch-and-Bound. $d$ is the dimension of Sparse PCA problem, and $k$ is the sparsity constant. We set the time limit to 3600 seconds for each method. Approximation errors between two methods, and speedup factors are reported. We note that the objective value for ArceneCov is scaled by $10^5$.

| Dataset | $d$ | $k$ | Branch-and-Bound | | Ours | | Error (%) | Speedup |
|---|---|---|---|---|---|---|---|---|
| | | | Time | Objective | Time(s) | Objective | | |
| CovColonCov | 500 | 3 | 8 | 1059.49 | 1.20 | 1059.49 | 0.00 | 6.67 |
| CovColonCov | 500 | 5 | 8 | 1646.45 | 1.08 | 1646.45 | 0.00 | 7.41 |
| CovColonCov | 500 | 10 | 3600 | 2641.23 | 42.93 | 2641.23 | 0.00 | 83.86 |
| CovColonCov | 500 | 15 | 3600 | 3496.6 | 666.95 | 3496.6 | 0.00 | 5.40 |
| LymphomaCov1 | 500 | 3 | 11 | 2701.61 | 1.32 | 2701.61 | 0.00 | 8.33 |
| LymphomaCov1 | 500 | 5 | 44 | 4300.5 | 1.51 | 4300.5 | 0.00 | 29.14 |
| LymphomaCov1 | 500 | 10 | 3600 | 6008.74 | 85.72 | 6008.74 | 0.00 | 42.00 |
| LymphomaCov1 | 500 | 15 | 3600 | 7628.66 | 324.6 | 7628.66 | 0.00 | 11.09 |
| Reddit1500 | 1500 | 3 | 75 | 946.12 | 3.68 | 946.12 | 0.00 | 20.38 |
| Reddit1500 | 1500 | 5 | 3600 | 980.97 | 6.86 | 980.97 | 0.00 | 524.78 |
| Reddit1500 | 1500 | 10 | 3600 | 1045.74 | 9.20 | 1045.74 | 0.00 | 391.30 |
| Reddit1500 | 1500 | 15 | 3600 | 1082.12 | 6.87 | 1082.12 | 0.00 | 524.02 |
| Reddit2000 | 2000 | 3 | 412 | 1311.36 | 8.70 | 1311.36 | 0.00 | 47.36 |
| Reddit2000 | 2000 | 5 | 3600 | 1397.36 | 9.03 | 1397.36 | 0.00 | 398.67 |
| Reddit2000 | 2000 | 10 | 3600 | 1523.82 | 13.18 | 1523.82 | 0.00 | 273.14 |
| Reddit2000 | 2000 | 15 | 3600 | 1605.48 | 9.10 | 1601.32 | 0.26 | 395.60 |
| LeukemiaCov | 3571 | 3 | 280 | 7.13 | 18.89 | 7.13 | 0.00 | 14.82 |
| LeukemiaCov | 3571 | 5 | 3600 | 10.29 | 18.64 | 10.29 | 0.00 | 193.13 |
| LeukemiaCov | 3571 | 10 | 3600 | 17.22 | 64.03 | 17.22 | 0.00 | 56.22 |
| LeukemiaCov | 3571 | 15 | 3600 | 21.87 | 168.47 | 21.87 | 0.00 | 21.37 |
| LymphomaCov2 | 4026 | 3 | 210 | 40.62 | 25.73 | 40.62 | 0.00 | 8.16 |
| LymphomaCov2 | 4026 | 5 | 144 | 63.66 | 26.75 | 63.66 | 0.00 | 5.38 |
| LymphomaCov2 | 4026 | 10 | 3600 | 78.29 | 293.42 | 69.27 | 11.52 | 12.27 |
| LymphomaCov2 | 4026 | 15 | 3600 | 93.46 | 1810.19 | 86.20 | 7.77 | 1.99 |
| ProstateCov | 6033 | 3 | 102 | 8.19 | 57.35 | 8.19 | 0.00 | 1.78 |
| ProstateCov | 6033 | 5 | 3600 | 12.92 | 54.63 | 12.92 | 0.00 | 65.90 |
| ProstateCov | 6033 | 10 | 3600 | 24.38 | 407.14 | 24.38 | 0.00 | 8.84 |
| ProstateCov | 6033 | 15 | 3600 | 34.98 | 1175.40 | 34.98 | 0.00 | 3.06 |
| ArceneCov | 10000 | 3 | 88 | 3.36 | 138.07 | 3.36 | 0.00 | 0.64 |
| ArceneCov | 10000 | 5 | 248 | 5.56 | 135.47 | 5.56 | 0.00 | 1.83 |
| ArceneCov | 10000 | 10 | 3600 | 10.81 | 138.56 | 10.81 | 0.00 | 25.98 |
| ArceneCov | 10000 | 15 | 3600 | 15.30 | 140.92 | 15.30 | 0.00 | 25.55 |

$k$, as the largest block size $d_0$ is set to 30, and the best solutions are typically found in blocks near this size.

From the data in Table 8, it is clear that our framework consistently obtains significant computational speedups across a range of settings. Notably, it achieves an average speedup factor of 13.52 for $k = 3$, with this factor increasing dramatically to over 153 for $k \geq 5$. An observation from the analysis is the relationship between $k$ and performance metrics: although the speedup factor increases with larger $k$, there is a corresponding rise in the average approximation errors. Specifically, for $k = 3$ and $k = 5$, the average errors remain exceptionally low, which are all zero, but they escalate to 1.47% for $k = 10$ and 1.05% for $k = 15$. This indicates a consistent pattern of increased approximation errors as $k$ grows, which is aligned with Theorem 3, despite the significant gains in speed.

## C.3 Empirical results when integrated with Chan's algorithm

In this section, we provide the detailed numerical test results of our framework when integrated with Chan's algorithm (previously summarized in Table 4), in Table 9. We provide statistics related to the

Table 7: Additional statistics regarding the best solution obtained by Algorithm 5 in Table 6. Denote by $\varepsilon^\star$ the threshold found that yields the best solution in Algorithm 5, $A$ the input covariance matrix, $A^{\varepsilon^\star}$ is thresholded matrix obtained by Algorithm 1 with input $(A, \varepsilon^\star)$.

| Dataset | k | Error (%) | $\varepsilon^\star / \|A\|_\infty$ | Zeros in $A^{\varepsilon^\star}$ (%) | Jaccard Index |
|---|---|---|---|---|---|
| CovColonCov | 3 | 0 | 0.75 | 99.988 | 1.00 |
| CovColonCov | 5 | 0 | 0.59 | 99.941 | 1.00 |
| CovColonCov | 10 | 0 | 0.58 | 99.926 | 1.00 |
| CovColonCov | 15 | 0 | 0.59 | 99.941 | 1.00 |
| LymphomaCov1 | 3 | 0 | 0.50 | 99.982 | 1.00 |
| LymphomaCov1 | 5 | 0 | 0.50 | 99.982 | 1.00 |
| LymphomaCov1 | 10 | 0 | 0.38 | 99.951 | 1.00 |
| LymphomaCov1 | 15 | 0 | 0.35 | 99.940 | 1.00 |
| Reddit1500 | 3 | 0 | 0.13 | 99.999 | 1.00 |
| Reddit1500 | 5 | 0 | 0.06 | 99.997 | 1.00 |
| Reddit1500 | 10 | 0 | 0.06 | 99.997 | 1.00 |
| Reddit1500 | 15 | 0 | 0.06 | 99.997 | 1.00 |
| Reddit2000 | 3 | 0 | 0.13 | 99.999 | 1.00 |
| Reddit2000 | 5 | 0 | 0.09 | 99.998 | 1.00 |
| Reddit2000 | 10 | 0 | 0.09 | 99.998 | 1.00 |
| Reddit2000 | 15 | 0.26 | 0.09 | 99.998 | 0.88 |
| LeukemiaCov | 3 | 0 | 0.41 | 99.998 | 1.00 |
| LeukemiaCov | 5 | 0 | 0.50 | 99.999 | 1.00 |
| LeukemiaCov | 10 | 0 | 0.50 | 99.999 | 1.00 |
| LeukemiaCov | 15 | 0 | 0.44 | 99.999 | 1.00 |
| LymphomaCov2 | 3 | 0 | 0.50 | 99.999 | 1.00 |
| LymphomaCov2 | 5 | 0 | 0.50 | 99.999 | 1.00 |
| LymphomaCov2 | 10 | 11.52 | 0.42 | 99.999 | 0.42 |
| LymphomaCov2 | 15 | 7.77 | 0.44 | 99.999 | 0.30 |
| ProstateCov | 3 | 0 | 0.75 | 99.999 | 1.00 |
| ProstateCov | 5 | 0 | 0.68 | 99.999 | 1.00 |
| ProstateCov | 10 | 0 | 0.69 | 99.999 | 1.00 |
| ProstateCov | 15 | 0 | 0.69 | 99.999 | 1.00 |
| ArceneCov | 3 | 0 | 0.75 | 99.999 | 1.00 |
| ArceneCov | 5 | 0 | 0.75 | 99.999 | 1.00 |
| ArceneCov | 10 | 0 | 0.75 | 99.999 | 1.00 |
| ArceneCov | 15 | 0 | 0.75 | 99.999 | 1.00 |

Table 8: Summary of average approximation errors and average speedup factors for each $k$, when integrated our framework with Branch-and-Bound algorithm. The standard deviations are reported in the parentheses.

| k | Avg. Error (Std. Dev) | Avg. Speedup (Std. Dev) |
|---|---|---|
| 3 | 0.00 (0.00) | 13.52 (15.12) |
| 5 | 0.00 (0.00) | 153.28 (203.17) |
| 10 | 1.47 (4.06) | 111.70 (141.81) |
| 15 | 1.05 (2.72) | 123.51 (210.55) |

best solution in Table 10, including the ratio of corresponding $\varepsilon^\star$ and $\|A\|_\infty$, the percentage of zeros in the matrix $A^{\varepsilon^\star}$, and the Jaccard index. We also summarized the average approximation errors and average speedup factors for with respect to $k$ in Table 11.

In Table 9, we once again demonstrate the substantial speedup improvements our framework achieves when integrated with Chan's algorithm. The results show a median approximation error of -0.38%, with the maximum approximation error not exceeding 0.28%, and a median speedup factor of 3.96. We observe that the speedup factor grows (linearly) as the dimension of the dataset goes up,

highlighting the efficacy of our framework, especially in large datasets. Notably, in all instances within the DorotheaCov dataset, the instances $k = 500, 1000, 2000$ in GLI85Cov and GLABRA180Cov datasets, the instances $k = 1000, 2000$ in ArceneCov dataset, our framework attains solutions with negative approximation errors, indicating that it consistently finds better solutions than Chan's algorithm alone. These instances provide tangible proof of our framework's potential to find better solutions, as highlighted in Remark 2.

In Table 10, the Jaccard index does not vary significantly with $k$, nor does its change align with the change of the approximation error. A possible reason is that Chan's algorithm is an approximation algorithm, and the Jaccard index may not fully capture the similarity between the solutions obtained and the true optimal solutions. As expected, since $d_0 = 2k$, the relative threshold value $\varepsilon^\star / \|A\|_\infty$ and the percentage of zeros decrease as $k$ increases.

In Table 11, we do not observe an increase in speedup factors as $k$ increases. This phenomenon can be attributed to the runtime of Chan's algorithm, which is $\mathcal{O}(d^3)$ and independent of $k$. Consequently, as predicted by Proposition 1, the speedup factors do not escalate with larger values of $k$. Additionally, unlike the trends noted in Table 8, the average errors do not increase with $k$. This deviation is likely due to Chan's algorithm only providing an approximate solution with a certain multiplicative factor: As shown in Theorem 3, our algorithm has the capability to improve this multiplicative factor at the expense of some additive error. When both $k$ and $d$ are large, the benefits from the improved multiplicative factor tend to balance out the impact of the additive error. Consequently, our framework consistently achieves high-quality solutions that are comparable to those obtained solely through Chan's algorithm, maintaining an average error of less than 0.55% for each $k$.

Finally, we note that Chan's algorithm is considerably more scalable than the Branch-and-Bound algorithm. The statistics underscore our framework's ability not only to enhance the speed of scalable algorithms like Chan's, but also to maintain impressively low approximation errors in the process.

Table 9: Comparison of runtime and objective values on real-world datasets, between Chan's algorithm and Algorithm 5 integrated with Chan's algorithm. $d$ is the dimension of Sparse PCA problem, and $k$ is the sparsity constant. We note that the objective value for GLI85Cov is scaled by $10^9$ and that for GLABRA180Cov is scaled by $10^8$. The error being negative means that our framework finds a better solution than the standalone algorithm.

| Dataset | $d$ | $k$ | Chan's algorithm | | Ours | | Error | Speedup |
|---|---|---|---|---|---|---|---|---|
| | | | Time | Objective | Time | Objective | | |
| ArceneCov | 10000 | 200 | 228 | 91.7 | 288 | 91.7 | 0.00 | 0.79 |
| ArceneCov | 10000 | 500 | 239 | 140.4 | 188 | 140.4 | 0.00 | 1.27 |
| ArceneCov | 10000 | 1000 | 249 | 190.4 | 349 | 190.8 | -0.21 | 0.71 |
| ArceneCov | 10000 | 2000 | 256 | 233.3 | 377 | 233.5 | -0.09 | 0.68 |
| GLI85Cov | 22283 | 200 | 2836 | 35.8 | 1660 | 35.7 | 0.28 | 1.71 |
| GLI85Cov | 22283 | 500 | 2728 | 38.8 | 1528 | 39.4 | -1.55 | 1.79 |
| GLI85Cov | 22283 | 1000 | 2869 | 39.9 | 1942 | 40.9 | -2.51 | 1.48 |
| GLI85Cov | 22283 | 2000 | 2680 | 42.3 | 2034 | 42.5 | -0.47 | 1.32 |
| GLABRA180Cov | 49151 | 200 | 35564 | 44.7 | 4890 | 44.6 | 0.11 | 7.27 |
| GLABRA180Cov | 49151 | 500 | 33225 | 60.4 | 4475 | 60.7 | -0.50 | 7.42 |
| GLABRA180Cov | 49151 | 1000 | 30864 | 70.5 | 4142 | 70.7 | -0.28 | 7.45 |
| GLABRA180Cov | 49151 | 2000 | 29675 | 78.1 | 4830 | 78.2 | -0.10 | 6.14 |
| Dorothea | 100000 | 200 | 246348 | 3.25 | 16194 | 3.34 | -2.77 | 15.21 |
| Dorothea | 100000 | 500 | 246539 | 4.94 | 17554 | 5.07 | -2.63 | 14.04 |
| Dorothea | 100000 | 1000 | 244196 | 6.23 | 16934 | 6.38 | -2.41 | 14.42 |
| Dorothea | 100000 | 2000 | 255281 | 7.18 | 17774 | 7.28 | -1.39 | 14.36 |

## C.4 EMPIRICAL RESULTS UNDER MODEL 1

In this section, we report the numerical results in Model 1 using Algorithm 3 with a threshold obtained by Algorithm 4. In Model 1, we set $E$ to have i.i.d. centered Gaussian variables with a standard deviation $\sigma = 0.1$ in its lower triangle entries, i.e., $E_{ij}$ for $1 \le i \le j$, and set $E_{ij} = E_{ji}$ for $i \ne j$ to

Table 10: Additional statistics regarding the best solution obtained by Algorithm 5 in Table 9. Denote by $\varepsilon^\star$ the threshold found that yields the best solution in Algorithm 5, $A$ the input covariance matrix, $A^{\varepsilon^\star}$ is thresholded matrix obtained by Algorithm 1 with input $(A, \varepsilon^\star)$.

| Dataset | k | Error (%) | $\varepsilon^\star / \|A\|_\infty$ | Zeros in $A^{\varepsilon^\star}$ (%) | Jaccard Index |
|---|---|---|---|---|---|
| ArceneCov | 200 | 0.00 | 0.38 | 99.974 | 0.82 |
| ArceneCov | 500 | 0.00 | 0.26 | 99.868 | 0.82 |
| ArceneCov | 1000 | -0.21 | 0.16 | 99.469 | 0.79 |
| ArceneCov | 2000 | -0.09 | 0.05 | 95.891 | 0.94 |
| GLI85Cov | 200 | 0.28 | 0.09 | 99.997 | 0.67 |
| GLI85Cov | 500 | -1.55 | 0.04 | 99.985 | 0.49 |
| GLI85Cov | 1000 | -2.51 | 0.02 | 99.962 | 0.62 |
| GLI85Cov | 2000 | -0.47 | 0.01 | 99.775 | 0.73 |
| GLABRA180Cov | 200 | 0.11 | 0.18 | 99.999 | 0.82 |
| GLABRA180Cov | 500 | -0.50 | 0.10 | 99.997 | 0.81 |
| GLABRA180Cov | 1000 | -0.28 | 0.05 | 99.990 | 0.82 |
| GLABRA180Cov | 2000 | -0.10 | 0.04 | 99.980 | 0.80 |
| DorotheaCov | 200 | -2.77 | 0.12 | 99.999 | 0.78 |
| DorotheaCov | 500 | -2.63 | 0.09 | 99.999 | 0.79 |
| DorotheaCov | 1000 | -2.41 | 0.06 | 99.999 | 0.76 |
| DorotheaCov | 2000 | -1.39 | 0.05 | 99.998 | 0.67 |

Table 11: Summary of average approximation errors and average speedup factors for each $k$, when integrated our framwork with Chan's algorithm. The standard deviations are reported in the parentheses.

| k | Avg. Error (Std. Dev) | Avg. Speedup (Std. Dev) |
|---|---|---|
| 200 | -0.60 (1.45) | 6.25 (6.63) |
| 500 | -1.17 (1.17) | 6.13 (5.96) |
| 1000 | -1.35 (1.28) | 6.02 (6.36) |
| 2000 | -0.51 (0.61) | 5.63 (6.31) |

make it symmetric. We generate 30 independent random blocks in $\widetilde{A}$, each of size 20, with each block defined as $M_i^\top M / 100$, where $M \in \mathbb{R}^{100 \times 20}$ has i.i.d. standard Gaussian entries (which implies that $u = 1$ in Model 1).

We run Algorithm 4 with $C = 1$, $\alpha = 0.7$, and $u = 1$, obtaining the threshold $\bar{\varepsilon}$ as the output. Subsequently, we execute Algorithm 3 with $\widetilde{A}$, $k \in \{2, 3, 5, 7, 10\}$, the Branch-and-Bound algorithm, and $\bar{\varepsilon}$. We denote the solution output by Algorithm 3 as $x_{\text{Ours}}$.

In Table 12, we compare Algorithm 3 integrated with Branch-and-Bound algorithm, with vanilla Branch-and-Bound algorithm. We report the optimality gap, speed up factor, and the value of $\bar{\varepsilon}$ outputted by Algorithm 4. The optimality gap is defined as:

$$\text{Gap} := \text{Obj}_{\text{BB}} - \text{Obj}_{\text{Ours}},$$

where $\text{Obj}_{\text{BB}} := x_{\text{BB}}^\top \widetilde{A} x_{\text{BB}}$, and $x_{\text{BB}}$ is the output of the Branch-and-Bound algorithm with input $(\widetilde{A}, k)$, and where $\text{Obj}_{\text{Ours}}$ is the objective value $y^\top \widetilde{A} y$ in Proposition 2.

To ensure reproducibility, we set the random seed to 42 and run the experiments ten times for each $k$.

From Table 12, we observe that the average optimality gap increases as $k$ grows. Additionally, $\bar{\varepsilon}$ remains relatively stable across different values of $k$, as the calculation of $\bar{\varepsilon}$ does not depend on $k$ at all. The optimality gap is much smaller than the predicted bound $3k \cdot \bar{\varepsilon}$, providing computational verification of the bound proposed in Proposition 2. The speedup factor is exceptionally high, often exceeding a thousand when $k \geq 5$.

Table 12: Summary of average optimality gaps, average speedup factors, and average threshold $\bar{\varepsilon}$ for each $k$, when integrated our framework with Branch-and-Bound algorithm. The standard deviations are reported in the parentheses. The time limit for Branch-and-Bound is set to 600 seconds.

| k | Avg. Gap (Std. Dev) | Avg. Spdup (Std. Dev) | Avg. $\bar{\varepsilon}$ (Std. Dev) |
|---|---|---|---|
| 2 | 0.29 (0.08) | 16.52 (12.18) | 0.96 (0.0016) |
| 3 | 0.35 (0.13) | 94.78 (78.52) | 0.96 (0.0030) |
| 5 | 0.48 (0.12) | 2010.02 (625.55) | 0.96 (0.0028) |
| 7 | 0.59 (0.14) | 1629.38 (869.91) | 0.96 (0.0040) |
| 10 | 0.64 (0.14) | 1900.05 (598.14) | 0.96 (0.0036) |

## C.5 IMPACTS OF PARAMETERS $d_0$ AND $\delta$ IN ALGORITHM 5

In this section, we discuss the impacts of parameters $d_0$ and $\delta$ in Algorithm 5. We showcase the impacts by comparing the approximation errors and speedups on certain datasets when integrating Algorithm 5 with Branch-and-Bound algorithm.

### C.5.1 IMPACT OF $d_0$

In Table 13, we summarize the numerical results when increasing $d_0$ from 30 to 45 and 55 in the LymphomaCov2 dataset. We observe a significant improvement on the approximation error, going from an average of 4.82% to 1.94% and 0.03%, at the cost of lowering the average speedup factor from 6.95 to around 4.

Table 13: Comparison of errors and speedups for different $d_0$ values across LymphomaCov2 dataset.

| Dataset | $d$ | $k$ | $d_0 = 30$ | | $d_0 = 45$ | | $d_0 = 55$ | |
|---|---|---|---|---|---|---|---|---|
| | | | Error | Speedup | Error | Speedup | Error | Speedup |
| LymphomaCov2 | 4026 | 3 | 0.0 | 8.16 | 0.0 | 8.29 | 0.0 | 8.62 |
| LymphomaCov3 | 4026 | 5 | 0.0 | 5.38 | 0.0 | 5.35 | 0.0 | 5.95 |
| LymphomaCov4 | 4026 | 10 | 11.52 | 12.27 | 0.0 | 1.84 | 0.0 | 1.00 |
| LymphomaCov5 | 4026 | 15 | 7.77 | 1.99 | 7.77 | 0.92 | 0.13 | 1.00 |
| Overall | | | 4.82 | 6.95 | 1.94 | 4.10 | 0.03 | 4.14 |

### C.5.2 IMPACT OF $\delta$

In Table 14, we summarize the numerical results when increasing $\delta$ from $0.01 \cdot \|A\|_\infty$ to $0.1 \cdot \|A\|_\infty$ and $0.15 \cdot \|A\|_\infty$ in the ProstateCov dataset. We observe that, by increasing $\delta$, the speedup factors dramatically improve, from an average of 19.90 to 38.41 and 99.49, at the potential cost of lowering the approximation error from 0 to 6.07%.

Table 14: Comparison of errors and speedups for different $\delta$ values across ProstateCov dataset.

| Dataset | $d$ | $k$ | $\delta = 0.01 \cdot \|A\|_\infty$ | | $\delta = 0.1 \cdot \|A\|_\infty$ | | $\delta = 0.15 \cdot \|A\|_\infty$ | |
|---|---|---|---|---|---|---|---|---|
| | | | Error | Speedup | Error | Speedup | Error | Speedup |
| ProstateCov | 6033 | 3 | 0.0 | 1.78 | 0.0 | 2.79 | 0.0 | 3.62 |
| ProstateCov | 6033 | 5 | 0.0 | 65.90 | 0.0 | 106.32 | 0.0 | 128.16 |
| ProstateCov | 6033 | 10 | 0.0 | 8.84 | 0.0 | 29.56 | 0.0 | 128.25 |
| ProstateCov | 6033 | 15 | 0.0 | 3.06 | 0.0 | 14.99 | 24.27 | 137.93 |
| Overall | | | 0.0 | 19.90 | 0.0 | 38.41 | 6.07 | 99.49 |

## D    EXTENSION TO NON-POSITIVE SEMIDEFINITE INPUT MATRIX

In this paper, we focus on solving SPCA, where the input matrix $A$ is symmetric and positive semidefinite, following conventions in the Sparse PCA literature (Chan et al., 2015; Amini & Wainwright, 2008; Berk & Bertsimas, 2019; Dey et al., 2022a;b). We remark that, our framework remains effective for symmetric and non-positive semidefinite inputs, provided that the algorithm $\mathcal{A}$ for (approximately) solving SPCA supports symmetric and non-positive semidefinite inputs.[2] In other words, if $\mathcal{A}$ is compatible with symmetric and non-positive semidefinite inputs, the assumption of $A$ being positive semidefinite can be removed in Theorems 3 and 4 and in Proposition 3 (stated in Model 1), and all theoretical guarantees still hold.

---

[2]Such algorithms do exist; for example, a naive exact algorithm that finds an eigenvector corresponding to the largest eigenvalue of submatrices $A_{S,S}$ for all subsets $S$ with $|S| \leq k$.

