# OpenReview forum: "Efficient Sparse PCA via Block-Diagonalization"
_ICLR.cc/2025/Conference — ICLR 2025 Poster_

### Official Review · Reviewer_fTk5 · 2024-10-29

**Soundness:** 3
**Presentation:** 2
**Contribution:** 3
**Rating:** 8
**Confidence:** 4

**Summary:**

This paper proposes to address the one-component sparse PCA (SPCA) problem by first approximating the target covariance matrix by a block-diagonal matrix (by thresholding its entries and permuting rows and columns), and then looking for solutions within each one of the blocks. These smaller SPCA problems associated with each block can be (exactly or approximately) solved by means of existing SPCA algorithms, and are expected to require (potentially much) less compute in comparison with the standard approach. The authors state and prove some results bounding the suboptimality of this scheme (in terms of objective function values) in terms of the infinity norm of the matrix approximation error, the sparsity level and the largest size among the obtained blocks, including in their analysis the case in which an approximate algorithm with additive and multiplicative errors is used. Simulation results are given, comparing the proposed scheme with two other approaches: exact solution by branch-and-bound, and approximate solution by Chan's algorithm (reference Chan et al., 2015 cited by the authors).

**Strengths:**

S1) The proposed scheme is simple and intuitively well-motivated.

S2) The theoretical results are also fairly simple and support the intuition.

S3) The analysis includes approximate algorithms, which have great practical relevance since the problem is computationally hard.

S4) The simulation results demonstrate a significant speedup in large problem instances, most often with only a small relative suboptimality gap. Hence, the proposed algorithm is clearly useful for practitioners.

**Weaknesses:**

W1) Some of the technical contents, including the problem formulation, are not properly formalized, with some important aspects completely omitted. Notably, it should be stated that matrix $A$ in (SPCA) is necessarily symmetric positive semidefinite, implying the same for the blocks in the block-diagonal approximation. The absence of this information and the fact that throughout the paper $A$ is simply referred to as an "input matrix" rather than a covariance matrix may mislead the reader into thinking that the problem is more general than it actually is.

W2) The presentation of the simulation results is somewhat superficial, focusing only on presenting and briefly commenting the two quantitative criteria used for comparison, without much discussion or insight into what is going on. Specifically:
- Separate values for the different used $k$ should be reported (see point W2 below).
- It would be interesting to report the threshold value $\varepsilon$ effectively chosen by the algorithm (relatively to the infinity norm of the input matrix), as well as the proportion of zero entries after the thresholding.
- It would also be interesting to compare the support of the solution obtained by the proposed scheme with that obtained by the baseline methods (e.g., using a Jaccard index).

W3) Reporting average results with respect to $k$ is not a reasonable choice in my opinion, as the statistics of the chosen metrics probably behave very differently for different values of $k$.

W4) As it is, I don't see the utility of Section 4.1. First, this model is not applied to any of the datasets used in the experiments. This leads one to suspect that in practice it is quite hard to come up with estimates of the parameters required by Algorithm 4. Second, the authors do not even generate synthetic data following such a model (which is one typical use of a model) in order to illustrate the obtained results. In my view results of this kind should be added, or else the contents of 4.1 should be moved to the appendices as they're not really important (or both).

W5) Though generally well-written, the paper lacks some clarity at times, and the notation is not always consistent/clear. In particular:
- The sentence "This result is non-trivial: while the support of an optimal solution could span multiple blocks, we theoretically show that there must exist a block that contains the support of an optimal solution, which guarantees the efficiency of our framework." seems to be contradicatory. I believe that the authors mean the following: *one could think* that the solution could span multiple blocks, but they show this is not true. The same goes for Remark 1.
- What is the difference between $A^\varepsilon$ and $\tilde{A}$ in Theorem 1? It seems that two different symbols are used to denote the same object.
- The constraint $\|x\|_0 \le k$ in the last problem that appears in the proof of Theorem 2 is inocuous, since the size of each block $\tilde{A}_i'$ is at most $k$ anyway. This should be commented.

**Questions:**

I suggest that the authors take the above stated weaknesses into account to improve their manuscript. Apart from this suggestion:

Q1) How far from the bound implied by Theorem 1 are the measured suboptimality gaps in practice?

Q2) How realistic and how restrictive is the bound on $d^{1-\alpha}$ of Proposition 1? How can it be interpreted? A discussion should be presented on this point, as it is not obvious.

Q3) How hard is it to extend your results to the multi-component case?

---

> ### Author Response · Authors · 2024-11-22
>
> Dear Reviewer,
>
> Thank you for taking the time to review our paper and provide thoughtful comments. We have outlined our responses and clarifications below. We hope that our efforts have addressed your concerns, and we kindly request that you consider increasing your scores.
>
> **Responses to Weaknesses:**
>
> 1. > Some of the technical contents, including the problem formulation, are not properly formalized, with some important aspects completely omitted. Notably, it should be stated that matrix in (SPCA) is necessarily symmetric positive semidefinite, implying the same for the blocks in the block-diagonal approximation. The absence of this information and the fact that throughout the paper is simply referred to as an "input matrix" rather than a covariance matrix may mislead the reader into thinking that the problem is more general than it actually is.
>
> **Response:** We thank you for pointing this out. In fact, our framework requires the input matrix $A$ to be symmetric. We have made the clarification on Page 3 in our revised manuscript. Other than that, our framework accommodates general symmetric input matrices, provided the subroutine used is compatible with such matrices. For PSD input matrices, our framework can be modified to achieve better approximation error. Specifically, if we define $d^\star = intdim(A, \epsilon)$, the additive error term in Theorem 3 decreases from $a\left(k, d^\star\right) + (1 + \frac{1}{m\left(k, d^\star\right)}) \cdot k\epsilon$ to $a\left(k, d^\star\right) + \frac{2k\epsilon}{m\left(k, d^\star\right)},$ using a similar proof to the one presented for Theorem 3. The modification is on Algorithm 2: instead of outputting the thresholded matrix blocks, we output the corresponding matrix blocks from the original input matrix $A$, which ensures the blocks are PSD. We are happy to add more discussions on this during the revision.
>
>
>
> 2. > The presentation of the simulation results is somewhat superficial, focusing only on presenting and briefly commenting the two quantitative criteria used for comparison, without much discussion or insight into what is going on. Specifically:
> > 1. Separate values for the different $k$ used should be reported (see point below).
> > 2. It would be interesting to report the threshold value $\epsilon$ effectively chosen by the algorithm (relatively to the infinity norm of the input matrix), as well as the proportion of zero entries after the thresholding.
> > 3. It would also be interesting to compare the support of the solution obtained by the proposed scheme with that obtained by the baseline methods (e.g., using a Jaccard index).
>
> **Responses:** We appreciate your suggestions. Regarding the values for separate $k$, we have reported and discussed the average speedup factors and average approximation errors in Appendices C.2 and C.3 of our original manuscript. The summary of results across different $k$ values was not included in the main text due to space constraints.
> We next report the relative threshold value $\epsilon / |A|$, the percentage of zero entries after the thresholding, and Jaccard index for selected datasets. We first present the results on the LymphomaCov1 dataset, comparing our Algorithm 5 with Branch-and-Bound algorithm and the vanilla Branch-and-Bound algorithm:
>
> |k|Error (%)|relative $\epsilon$|zero percentage (%)|jaccard index|
> |--|---------|-------------------|-------------------|-------------|
> |3|0|0.313|99.919|1.000|
> |5|0|0.315|99.919|1.000|
> |10|1.64|0.305|99.910|0.333|
> |15|0.46|0.307|99.912|0.875|
>
> From the table above, it is evident that the change of Jaccard index aligns with the change of the approximation error. Additionally, the relative threshold value $\epsilon / |A|$ and the percentage of zeros remain relatively stable across $k$, as the largest block size $d_0$ is set to 40, and the best solutions are typically found in blocks near this size.
> We next present results on the GLABRA180Cov dataset, comparing our Algorithm 5 with Chan’s algortihm and the vanilla Chan’s algorithm:
>
> |k|Error (%)|relative $\epsilon$|zero percentage (%)|jaccard index|
> |--|---------|-------------------|-------------------|-------------|
> |200|1.12|0.187|99.999|0.794|
> |500|0.33|0.101|99.997|0.815|
> |1000|0.71|0.062|99.992|0.792|
> |2000|0.9|0.047|99.985|0.771|
>
> For Chan’s algorithm, the Jaccard index does not vary significantly with $k$, nor does its change align with the change of the approximation error. A possible reason is that Chan’s algorithm is an approximation algorithm, and the Jaccard index may not fully capture the similarity between the solutions obtained and the true optimal solutions. As expected, since $d_0 = 2k$, the relative threshold value $\epsilon / |A|$ and the percentage of zeros decrease as $k$ increases.
> We are happy to conduct more experiments and add discussions on this during the revision.

---

> > ### Author Response · Authors · 2024-11-22
> >
> > 3. > Reporting average results with respect to $k$ is not a reasonable choice in my opinion, as the statistics of the chosen metrics probably behave very differently for different values of $k$.
> >
> > **Responses:** Please see our responses to the question above.
> >
> > 4. > As it is, I don't see the utility of Section 4.1. First, this model is not applied to any of the datasets used in the experiments. This leads one to suspect that in practice it is quite hard to come up with estimates of the parameters required by Algorithm 4. Second, the authors do not even generate synthetic data following such a model (which is one typical use of a model) in order to illustrate the obtained results. In my view results of this kind should be added, or else the contents of 4.1 should be moved to the appendices as they're not really important (or both).
> >
> > **Responses:** We thank you for your feedback regarding Section 4.1. Below, we clarify the utility of this section and how it supports our proposed framework:
> >
> > 1.*Purpose and Practical Utility*.  This section serves as a theoretical justification for our proposed framework, demonstrating that $\epsilon$ is not always required as an input, as it can often be efficiently estimated in some statistical models. To be more specific, a crucial aspect of our framework involves determining an appropriate threshold $\epsilon$ for Algorithm 3. Section 4.1 introduces a model that provides a structured way to estimate $\epsilon$ in many practical cases, as discussed on lines 344 - 347.
> >
> > 2.*Ease of Estimating Parameters*.  In practice, these parameters ($C$, $\alpha$, and $u$) are often easy to estimate when prior information about the block structure of the data is available: (i) $C$ and $\alpha$ can typically be inferred from prior knowledge of the block structure, especially when certain estimate is available; (ii) $u = 1$ when the noise $E$ follows a Gaussian distribution, which is a common assumption in many real-world scenarios.
> >
> > 3.*Additional Numerical Tests*.  Please see the following table for the additional numerical results conducted in Model 1. We set $E$ to be i.i.d. centered Gaussian variables with a standard deviation $\sigma = 0.1$ in its lower triangle entries, i.e., $E_{ij}$ for $1\le i\le j$, and set $E_{ij} = E_{ji}$ for $i\ne j$ to make it symmetric. We run Algorithm 4 with $C = 1$, $\alpha = 0.7$, and $u = 1$, obtaining the threshold $\bar{\epsilon}$ as the output. We use the Branch-and-Bound algorithm and $\bar{\epsilon}$ in Algorithm 3. We generate 30 independent random blocks in $\widetilde{A}$, each of size 20, with each block defined as $M_i^\top M / 100$, where $M \in \mathbb{R}^{100 \times 20}$ has i.i.d. standard Gaussian entries (which implies that $u=1$ in Model 1).
> > In the following table, we compare our Algorithm 3 integrated with Branch-and-Bound algorithm and vanilla Branch-and-Bound algorithm. We report the optimality gap, speed up factor, and the value of $\bar \epsilon$ outputted by Algorithm 4. The optimality gap is defined as $Obj_{BB} - Obj_{Ours}$, where $Obj_{BB}:= x_{BB}^\top \widetilde A x_{BB}$, and $x_{BB}$ is the output of the Branch-and-Bound algorithm with input $(\widetilde A, k)$; $Obj_{Ours}$ is $y^\top \widetilde A y$ in Proposition 2.
> >
> > |k|Avg. Gap (Std. Dev)|Avg. Spdup (Std. Dev)|Avg. $\bar\epsilon$ (Std. Dev)|
> > |--|-------------------|---------------------|-----------------------------|
> > |2|0.30 (0.10)|17.95 (13.28)|0.96 (0.01)|
> > |3|0.36 (0.12)|91.32 (66.10)|0.96 (0.01)|
> > |5|0.48 (0.13)|1856.68 (781.22)|0.96 (0.01)|
> > |7|0.59 (0.14)|1315.56 (837.27)|0.96 (0.01)|
> > |10|0.65 (0.13)|1837.19 (573.64)|0.96 (0.01)|
> >
> > From the table above, we observe that the average optimality gap increases as $k$ grows. Additionally, $\bar{\epsilon}$ remains relatively stable across different values of $k$, as the calculation of $\bar \epsilon$ does not depend on $k$ at all. The optimality gap is much smaller than the predicted bound $4k \cdot \bar{\epsilon}$, providing computational verification of the bound proposed in Proposition 2. The speedup factor is exceptionally high, often exceeding a thousand when $k > 5$. We have included the numerical results in Appendix C.4 in our revised paper.

---

> > > ### Author Response · Authors · 2024-11-22
> > >
> > > 5. > Though generally well-written, the paper lacks some clarity at times, and the notation is not always consistent/clear:
> > > > 1. The sentence "This result is non-trivial: while the support of an optimal solution could span multiple blocks, we theoretically show that there must exist a block that contains the support of an optimal solution, which guarantees the efficiency of our framework." seems to be contradicatory. I believe that the authors mean the following: one could think that the solution could span multiple blocks, but they show this is not true. The same goes for Remark 1.
> > > > 2. What is the difference between $A^{\epsilon}$ and $\bar A$ in Theorem 1? It seems that two different symbols are used to denote the same object.
> > > > 3. The constraint $|x|_0\le k$ in the last problem that appears in the proof of Theorem 2 is inocuous, since the size of each block is at most $k$ anyway. This should be commented.
> > >
> > > **Responses:** We thank you for the helpful advice. We appreciate it for the clarification on the sentence. However, we would like to clarify that we did not show that, for *any* optimal solution to SPCA, the solution must stay in one block; instead, we show that, there must *exist* one optimal solution whose support entirely lies in one block. We have modified this sentence to “This result is non-trivial: while the support of optimal solutions might span multiple blocks, we prove that there always exists an optimal solution whose support is contained within a single block, ensuring the efficiency of our framework.” and we have also modified Remark 1 to add clarity in our revised paper.
> > >
> > > Regarding the different notation in Theorem 1, this was a typo - $\bar A$ should be replaced by $A^{\epsilon}$, and we have fixed that in our revised manuscript.
> > >
> > > Finally, concerning the redundant constraint, our intention was to show that the last problem is exactly the maximum of $\text{OPT}_i$, thereby proving the theorem. We have added clarification to the proof to avoid further confusion, as highlighted on Page 16.
> > >
> > >
> > > **Responses to questions:**
> > > 1. > How far from the bound implied by Theorem 1 are the measured suboptimality gaps in practice?
> > >
> > > **Response:** Our numerical tests show that the measured gaps are much smaller than the predicted suboptimality gap in Theorem 1. Examples can be found in our replies to your concern regarding the second weakness of the paper, along with table 6 and table 10 in our paper. Notably, the best threshold chosen in Algorithm 5 is often a large proportion of $|A|_{\infty}$, yet 80% of the instances yield an approximation error less than 2%.
> > >
> > > 2. > How realistic and how restrictive is the bound on $d^{1-\alpha}$ of Proposition 1? How can it be interpreted? A discussion should be presented on this point, as it is not obvious.
> > >
> > > **Response:** We could not find $d^{1-\alpha}$ in Proposition 1, and we assume that you are referring to Proposition 2. We also realized that there was a typo in the inequality in Proposition 2, and the correct inequality should be $d^{1-\alpha} > C_0 \cdot (C+1) \cdot u \log(8C+8)$. We will fix this in our revised manuscript.
> > >
> > > We believe that the bound $d^{1-\alpha} > C_0 \cdot (C+1) \cdot u \log(8C+8)$ is not restrictive. It only requires that $d^{1-\alpha}$ is greater than or equal to a constant multiple of an estimated $u$, which is reasonable for large-scale SPCA problems. For instance, when $\alpha = 0.5$, $d = 10000$, $C = 1$, $C_0 = 18$, and $u = 1$ (noting that $u = 1$ when $E$ is Gaussian), the inequality holds.
> > >
> > > We have added a brief remark on this point, on Page 8 in our revised manuscript.
> > >
> > > 3. > How hard is it to extend your results to the multi-component case?
> > >
> > > **Response:** The difficulty lies in extending Theorem 2 to the multi-component case. In this case, we are not clear if there exists an optimal solution whose support lies entirely in a block. We leave it for future work.

---

> > > > ### Comment · Reviewer_fTk5 · 2024-11-22
> > > >
> > > > I appreciate the effort of the authors in carefully and convincingly responding to my remarks and questions.
> > > > As a result I have changed my rating.

---

> > > > > ### Author Response · Authors · 2024-11-22
> > > > >
> > > > > Thank you very much for your reply, and for raising the score. We are happy that we have clarified all your concerns. We will add more related discussion and additional experimental results during revision.

---

### Official Review · Reviewer_uKtp · 2024-11-01

**Soundness:** 3
**Presentation:** 2
**Contribution:** 2
**Rating:** 6
**Confidence:** 3

**Summary:**

This paper presents a block-diagonal-based approach to speed up Sparse PCA, preserving an average approximation error. The authors provide mathematical proof for this algorithm, and extensive testing on real-world datasets demonstrates consistent improvements that align with theoretical predictions.

**Strengths:**

1. This paper is the first to apply a block-diagonal-based method to the Sparse PCA problem, enhancing the balance between runtime efficiency and average approximation error.
2. The authors present theoretical analysis, providing guarantees for both approximation error and runtime complexity.
3. This approach is evaluated across many real-world datasets, demonstrating consistent improvements in runtime and approximation error.

**Weaknesses:**

1. In your algorithm, you noted that the input matrix could be solved using $p$ sub-matrices, but you did not provide a method to determine $p$ or explain the relationship of $d_i$ (besides stating that $\sum d_i = d$).
2. There is not much discussion about the impact of hyperparameters.
3. In your notation, $p$ in $p$-norm should not represent the same value as $p$ in diag($A_1, \cdots, A_p$).
4. Some descriptions in your Theorem, Proposition, and Proof are not rigorous.
5. Some proofs are not completely correct.

**Questions:**

1. In your algorithm, you noted that the input matrix could be solved using $p$ sub-matrices. Could you please give more details on determining $p$ or explaining the relationship of $d_i$ (besides stating that $\sum d_i = d$)?
2. Could you explain why $|| E ||_\infty$ is the estimation in Model 1?
3. In algorithm 4, why $m=\lfloor (2C+2)d^{1+\alpha}  \rfloor$?
4. I'm curious about the inequality in Line 71. Can you prove it?
5. The proof of Theorem 1 is incomplete. Could you complete it?
6. In the proof of Theorem 3, how do you come up with $k || A-A^\epsilon ||_\infty \leq \epsilon $?
7. Could you prove why the g() function in Eq.(2) is convex?
8. Do you think the inequality in Line 41-45 is too relaxed?
9. Some descriptions are not rigorous. For example, "an absolute constant $c > 0$". Apparently, an absolute value is greater than 0 in this case.
10. Including additional details in the proof of Theorem 4 would improve clarity. For instance, explaining how the running time is derived and whether the relaxation is reasonable would be helpful.
11. I noticed that you frequently use expressions like $ \max \max $ or $ \inf \inf \inf $ in your proof, which appears unprofessional. Could you consider rephrasing these using constraints for clarity?
12. Some sentences are too long for readers to understand. For example, Line 51-53. Could you rephrase them?

---

> ### Author Response · Authors · 2024-11-22
>
> Dear Reviewer,
>
> We are grateful for your detailed feedback and helpful suggestions. Please find our responses and clarifications below.
>
> **Responses to weaknesses:**
>
> 1. > In your algorithm, you noted that the input matrix could be solved using $p$ sub-matrices, but you did not provide a method to determine $p$ or explain the relationship of $d_i$ (besides stating that $\sum d_i = d$).
>
> **Response:** We thank the reviewer for highlighting this concern. However, in our paper, it is not necessary to know $p$ or the specific values of $d_i$ prior to running the algorithms, as these quantities become clear during execution. More specifically, $p$ simply represents the number of blocks obtained after running Algorithm 2, and $d_i$ denotes the dimensions of these blocks. Both $p$ and $d_i$ are determined automatically and become evident upon completion of Algorithm 2.
>
>
> 2. > There is not much discussion about the impact of hyperparameters.
>
> **Response:** Except for Algorithm 5, our framework does not include any hyperparameters (we assume that the parameters are known beforehand in Algorithm 4). We have discussed the hyperparameters $d_0$ and $\delta$ in Algorithm 5 in Appendix C, on Page 22 (Page 24 in our revised manuscript). In summary, increasing $d_0$ yields a better solution but also increases the runtime of our framework. Setting $\delta$ to a moderate value that scales with $|A|_{\infty}$ has minimal impact on solution quality and can speed up the framework compared to using a very small universal constant for $\delta$.
>
> 3. > In your notation, $p$ in $p$-norm should not represent the same value as in $diag(A_1, A_2, \dots, A_p)$. Some descriptions in your Theorem, Proposition, and Proof are not rigorous. Some proofs are not completely correct.
>
> **Response:** In fact, $p$ in $p$-norm is used only for vectors, while in $diag(A_1, A_2, \dots, A_p)$ the integer $p$ denotes the total number of blocks in matrices. To avoid further confusion, we have changed $p$-norm to $q$-norm on Page 4 in our revised manuscript.
>
> We believe that we have now provided rigorous statements and proofs in our revised manuscript, addressing your comments below. Additionally, we are happy to clarify any points that may still be unclear to you.
>
> **Responses to questions:**
>
> 1. > Could you explain why $|E|_{\infty}$ is the estimation in Model 1?
>
> **Response:** The intuition for estimating $|E|_{\infty}$ is that we want to obtain an estimate of $A$ in Model 1, and then by our denoising algorithm Algorithm 1, we can get rid of the impact of $E$ and find a high quality approximate solution to SPCA with input $(A, k)$.
>
> 2. > In algorithm 4, why $m = \lceil (2C+2) d^{1+\alpha} \rceil$?
>
> **Response:** We choose this particular value as we make use of the analysis proposed in Comminges et al. (2021), where the idea is to make sure there are sufficient random variables in each block to make accurate estimation. The details are left in Proof of Proposition 3 on Page 17 - 18.
>
> 3. > I'm curious about the inequality in Line 71. Can you prove it?
>
> **Response:** We cannot find any inequality on Line 71, as it is Figure 1 in our paper introducing our proposed framework. However, we are happy to prove it if you could clarify which inequality you are referring to.
>
> 4. > The proof of Theorem 1 is incomplete. Could you complete it?
>
> **Response:** We respectfully disagree with this assessment. We believe that we have proved the desired result stated in Theorem 1. However, we are happy to provide further clarification if you could indicate which part of the proof is not clear.
>
> 5. > In the proof of Theorem 3, how do you come up with $k|A - A^\epsilon| \le \epsilon$?
>
> **Response:** We thank the reviewer for spotting this typo. What we meant is $\epsilon \cdot k$ on the right-hand-side, and we have corrected it on Page 16 in our revised manuscript. Note that we indeed use the correct inequality in the last inequality in this proof.
>
> 6. > Could you prove why the g() function in Eq.(2) is convex?
>
> **Response:** We assume in Proposition 1 that the function $g(k,d)$ is convex with respect to $d$.
>
> 7. > Do you think the inequality in Line 41-45 is too relaxed?
>
> **Response:** We cannot find any inequality on lines 41–45, as this part is focused on the literature review. However, we are happy to provide further clarification if the reviewer could specify which inequality they are referring to.
>
> 8. > Some descriptions are not rigorous. For example, "an absolute constant $c>0$". Apparently, an absolute value is greater than 0 in this case.
>
> **Response:** What we intended to emphasize is that such a constant is a universal constant, independent of other factors, such as the dimension of the problem. It does not refer to absolute values.

---

> ### Author Response · Authors · 2024-11-22
>
> 9. > Including additional details in the proof of Theorem 4 would improve clarity. For instance, explaining how the running time is derived and whether the relaxation is reasonable would be helpful.
>
> **Response:** We have made clarifications in Theorem4, as well as included the additional details in the proof of Theorem 4 in our revised manuscript. The high level idea of obtaining the running time is that, the running time for each iteration is upper bounded by $\mathcal{O}\left( \left\lceil \frac{d}{d_0} \right\rceil \cdot g\left(k,d_0\right) + d^2\right)$. This is clear due to the fact that $g$ is convex and the trivial fact that $g(k, 0) = 0$, and the fact that the call of an algorithm $\mathcal{A}$ would be executed only if the intrinsic dimension is less than or equal to $d_0$. Since the total number of iterations is upper bounded by $\mathcal{O}(\log(|A| / \delta))$, we can obtain the desired running time. For the approximation error,  we directly make use of Theorem 3.
>
> 10. > I noticed that you frequently use expressions like $\max\max\max$ or $\inf\inf\inf$ in your proof, which appears unprofessional. Could you consider rephrasing these using constraints for clarity?
>
> **Response:** Our intention is to remain consistent with the expressions used in Comminges et al. (2021). This is a common mathematical notation in the literature, as each $\max$ or $\inf$ corresponds to taking the maximum or infimum with respect to different variables, indices, or distributions.
>
> 11. > Some sentences are too long for readers to understand. For example, Line 51-53. Could you rephrase them?
>
> **Response:** We are happy to rephrase the sentence in our revised manuscript. The intention is to introduce that there are three types of algorithms: Some algorithms are fast but yield sub-optimal solutions, others provide high-quality solutions at the cost of greater computational complexity, and a few achieve both efficiency and accuracy but only under specific statistical assumptions. We have modified the sentence in our revised manuscript, as highlighted on Page 1.

---

> ### Comment · Reviewer_uKtp · 2024-11-27
>
> Thank you for your response. Most of my questions are answered.
>
> I’m curious about the inequality in Line 771. Can you provide a proof for it? (Question 3)
>
> For the proof of Theorem 1, I think it would be clearer if the author could state $||A-A^\epsilon||_\infty \le 2\epsilon$.

---

> > ### Author Response · Authors · 2024-11-27
> >
> > Thank you for your feedback. We are happy to address your additional concerns in the following.
> >
> > First, we would like to clarify that, we did not intend to prove anything on Line 771 (in the original submission). Instead, this inequality gives a formal definition for what we mean by an approximate algorithm $\mathcal{A}$ having both an additive error $a(k,d)$ and a multiplicative factor $m(k,d)$. Specifically, it states that this algorithm finds a solution with an objective value that is at least a multiple of $1/m(k,d)$ of the true optimum reduced by $a(k,d)$.
> >
> > Second, we have added the fact that $||A - A^{\epsilon}||_{\infty} \le \epsilon$ in the proof of Theorem 1 to our revised manuscript. Note that the right-hand-side should be $\epsilon$ instead of $2\epsilon$.
> >
> > We hope that we have addressed all your concerns, and we would be grateful if you could consider increasing your scores.

---

### Official Review · Reviewer_vGQN · 2024-11-03

**Soundness:** 3
**Presentation:** 3
**Contribution:** 3
**Rating:** 8
**Confidence:** 2

**Summary:**

The paper introduces a new framework for efficiently approximating Sparse Principal Component Analysis (Sparse PCA) by transforming the input covariance matrix into a block-diagonal structure, allowing for computationally manageable sub-problems.  The proposed method involves three key steps: creating a block-diagonal approximation of the matrix, solving Sparse PCA for each block, and then reconstructing an approximate solution for the entire matrix. By focusing on smaller blocks, this approach achieves substantial speedups with minimal loss in accuracy. The framework is adaptable, integrating well with existing Sparse PCA algorithms and reducing overall complexity theoretical and emperically.

**Strengths:**

The idea is well-motivated, and the problem is relevant to the community. Despite the NP-hardness of sparse PCA (SPCA), the authors propose addressing it through matrix block diagonalization. This framework demonstrates advantages in time complexity over traditional methods, both theoretically and empirically. Additionally, the authors discuss how to determine the appropriate SNR threshold, $\epsilon$, within a statistical model $A = \widetilde{A} + E $ using the proposed algorithm.

**Weaknesses:**

The authors investigate the recovery of the first individual eigenvector and ensure correctness by establishing an upper bound on the gap between the corresponding eigenvalues in Theorem 1. However, the situation changes when considering the principal subspaces of the covariance matrix $\Sigma$, which are spanned by sparse leading eigenvectors. When leading eigenvalues are identical or close to each other, individual eigenvectors may become unidentifiable. Could the analysis in Theorem 1 be extended to handle the aformentioned case?

**Questions:**

See above.

---

> ### Author Response · Authors · 2024-11-22
>
> Dear Reviewer,
>
> We are grateful for your detailed feedback. Please find our responses and clarifications below.
>
> **Responses to Weaknesses:**
>
> 1. > The authors investigate the recovery of the first individual eigenvector and ensure correctness by establishing an upper bound on the gap between the corresponding eigenvalues in Theorem 1. However, the situation changes when considering the principal subspaces of the covariance matrix $\Sigma$, which are spanned by sparse leading eigenvectors. When leading eigenvalues are identical or close to each other, individual eigenvectors may become unidentifiable. Could the analysis in Theorem 1 be extended to handle the aformentioned case?
>
> **Response:** Thank you for raising this insightful question. We believe that the issue you mentioned does not affect any of our theorems. In essence, Theorems 1 and 2 do not rely on finding an optimal solution; rather, they are universally true properties of the optimal solutions to SPCA, independent of the specific method used to solve SPCA. For Theorems 3 and 4, we require only that an approximate algorithm achieves a certain approximation ratio. As long as the algorithm satisfies the stated approximation bound for the input, the theorems hold. For instance, Chan’s algorithm can provide such an approximate solution for any input matrix $A$, including the scenario you described with nearly identical leading eigenvalues.

---

### Official Review · Reviewer_ZrrB · 2024-11-04

**Soundness:** 3
**Presentation:** 3
**Contribution:** 2
**Rating:** 5
**Confidence:** 4

**Summary:**

This paper solves approximate Sparse PCA with matrix block-diagonalization. The proposed algorithm has 3 steps:
approximate a given matrix by a block diagonal,
solve separate problems and then, combine them.

The matrix preparation step re-arranges the rows and columns so that most of the energy of the entries lies in a block diagonal structure and zeros out the rest of the entries. This part is what I think is the hard part to compute and the hardness of this approximation ( possibly harder than Sparse PCA to start with) is one confusing thing.

The central theoretical innovation of the paper is the following structural property of an SPCA: The support of the optimal solution must always be contained within one block.  I think this is a cool observation (but not some deep theorem) and I have a few questions about it that I discuss later.

The combination of solutions, after this observation is realized, is clear: you only pick the solution from the correct sub-block, since there are no interaction terms to worry about across blocks. This is cool, and all subsequent running time results and combinations with other algorithms are clear.

Edit after reading replies and discussion: I think the authors sort of addressed my concern, I give a small upgrade in my score

**Strengths:**

The paper has a cool observation and I think is well written.
I have some concerns about the complexity of the approximation of a matrix by a block diagonal that I would really like clarified.
The experimental section is reasonable some it misses additional experiments that I would like to see.

**Weaknesses:**

I have issues with the complexity of the block-diagonal matrix approximation and also with the experimental section, as I clarify below.

Also some minor typos here:
Definition 1, I assume Aij and \tilde{A}_ij are the entries of the matrix, would be good to mention. I know its mentioned later in additional notation but would be better to include in this definition.

typo: On the other hand, there also exist a number of algorithms that takes* polynomial runtime

typo: The axies* are indices of A;

typo: th3, positive integer, and donote*

**Questions:**

q1) I was very confused about the matrix preparation part of the algorithm (i.e. best approximation by block -diagonal). I would think that finding the best block-diagonal matrix that approximates a given matrix A (using any reasonable metric of distance) is NP-hard. Are the authors claiming to achieve this in polynomial time for any matrix using their L_inf distance?
I was a little lost on the notation and didn't understand where the search over all permutations of rows and columns is addressed.
Please clarify the computational complexity of this problem and how the algorithm gets around checking all row and column permutations.


q2) I do not see why theorem 2 is so challenging? I think its a cool observation but I think of it simply as: There is no reason for the principal component to put any energy on positions where the off-diagonal entries are zero, since it gets no benefit from that. Therefore it will pick the block with the most energy and put all its l2 mass there. For example if a matrix is just diagonal, it will put all its energy on the largest diagonal entry in absolute value.

q3) Theorem 2 doesn't really use sparsity anywhere right? Its really a result about PCA for block-diagonal matrices, or do i misunderstand something?

Remark: Approximating matrices with block-diagonal is indeed a very interesting topic.See this interesting question and answer by Gowers on discovering near-block diagonal structure of a matrix by re-ordering and its connections to combinatorics:
https://mathoverflow.net/questions/68041/showing-block-diagonal-structure-of-matrix-by-reordering
Another flavor is the "Bandwidth minimization problem". There instead of blocks one tries to re-arrange columns and rows so that all the non-zeros are on a few bands around the diagonal. This is known to be np-hard even for 3 bands, see:
Complexity Results for Bandwidth Minimization
M. R. Garey, R. L. Graham, D. S. Johnson, D. E. Knuth

q4) On experimental evaluation.
Why don't the authors compare to other algorithms as done in Papailiopoulos et al. ICML 2013 and the algorithms compared from that paper? There is (FullPath) of (d’Aspremont et al., 2007b) and
the truncated power method (TPower) of (Yuan & Zhang, 2011). I believe Tpower is usually the fastest in practice.
I think the Chan paper may have the best theoretical performance but it's not clear how it performs in practice. I did not see any performance plots in that paper.

I'm willing to improve my score if the authors manage to convince me on these concerns.

---

> ### Author Response · Authors · 2024-11-22
>
> Dear reviewer,
>
> Thank you very much for your comments and suggestions. Please find below our responses and clarifications. We believe we have addressed all your concerns and would appreciate it if you could consider raising your scores.
>
> **Responses to Weaknesses:**
>
> 1. > I was very confused about the matrix preparation part of the algorithm (i.e. best approximation by block -diagonal). I would think that finding the best block-diagonal matrix that approximates a given matrix A (using any reasonable metric of distance) is NP-hard. Are the authors claiming to achieve this in polynomial time for any matrix using their L_inf distance? I was a little lost on the notation and didn't understand where the search over all permutations of rows and columns is addressed. Please clarify the computational complexity of this problem and how the algorithm gets around checking all row and column permutations.
>
> **Response:** We thank you for raising this question. While finding the best block-diagonal approximation to a matrix $A$ is a hard problem, we don’t need to find the best one. Instead, we only need to find a matrix approximation that is $\epsilon$ away (after permutation) under L_inf distance, for some moderate $\epsilon > 0$. It can be identified in $O(d^2)$ using Algorithm 1 and Algorithm 2. Although such approximation might be far away from the best block-diagonal approximation, applying Algorithm 3 to this approximation matrix yields desired computational speedups and approximation guarantees.
>
>
> 2. > I do not see why theorem 2 is so challenging? I think its a cool observation but I think of it simply as: There is no reason for the principal component to put any energy on positions where the off-diagonal entries are zero, since it gets no benefit from that. Therefore it will pick the block with the most energy and put all its l2 mass there. For example if a matrix is just diagonal, it will put all its energy on the largest diagonal entry in absolute value.
>
> **Response:** We thank you for highlighting this insightful high-level intuition behind the result. To the best of our knowledge, this specific result has not been previously developed in the literature. We conduct a rigorous proof by contradiction on pages 14–15 (and on Page 16 in our revised manuscript) in the Appendix to prove it formally. We also believe the implications of this result are non-trivial, as it plays a key role in the development of our framework, as outlined in Remark 1 on Page 6.
>
>
>
>
> 3. > Theorem 2 doesn't really use sparsity anywhere right? Its really a result about PCA for block-diagonal matrices, or do i misunderstand something?
>
> **Response:**  In Theorem 2, we develop a more general result that applies to both PCA and SPCA, depending on the values of $k$ and $d$. In the special case where $k = d$, we show that there exists an optimal solution to PCA that lies within one of the block sub-matrices. For the case where $k < d$, we demonstrate that finding a $k$-sparse optimal solution to SPCA is equivalent to solving SPCA within each block.
>
> 4. > On experimental evaluation. Why don't the authors compare to other algorithms as done in Papailiopoulos et al. ICML 2013 and the algorithms compared from that paper? There is (FullPath) of (d’Aspremont et al., 2007b) and
> the truncated power method (TPower) of (Yuan & Zhang, 2011). I believe Tpower is usually the fastest in practice. I think the Chan paper may have the best theoretical performance but it's not clear how it performs in practice. I did not see any performance plots in that paper.
>
> **Response:** We thank you for the feedback. Our framework is mainly designed to speed up SPCA algorithms that come with approximation guarantees. TPower, while efficient in practice, lacks such guarantees and may not converge to global optima. This was the primary reason for not including it in our initial comparisons.
>
> To address your concerns, we conducted additional experiments with FullPath and TPower. Specifically, we run three sets of experiments on the DorotheaCov dataset: (1) vanilla TPower, (2) Algorithm 5 with FullPath, and (3) Algorithm 5 with Chan’s algorithm. Table below shows the objective values identified by each approach for different values of $k$. As you can see, TPower only identifies sub-optimal solutions with objective values ~ 0.2, yet both Alg 5 w/ FullPath and Alg 5 w/ Chan’s obtain solutions with much higher objective values. We didn’t include vanilla FullPath and Chan’s algorithms in this table since they have not outputted a solution after 24 hours. In contrast, Alg 5 w/ FullPath and Alg 5 w/ Chan’s output solutions in 4 - 5 hours.
> | k| 3| 4| 5| 6| 7| 8| 9 |10|
> |------------------|-------|-------|-------|-------|-------|-------|-------|-------|
> | Tpower|0.201 | 0.207 | 0.209 | 0.209 | 0.210 | 0.211 | 0.213 | 0.215 |
> | Alg 5 w/ FullPath|0.273|0.343 | 0.401 | 0.457 | 0.502 | 0.544 | 0.592 | 0.631 |
> | Alg 5 w/ Chan’s| 0.272| 0.343 | 0.401 | 0.457 | 0.502 | 0.543 | 0.592 | 0.628 |

---

> > ### Author Response · Authors · 2024-11-22
> >
> > > There are many typos.
> > > 1. Definition 1, I assume Aij and \tilde{A}_ij are the entries of the matrix, would be good to mention. I know its mentioned later in additional notation but would be better to include in this definition.
> > > 2. typo: On the other hand, there also exist a number of algorithms that takes* polynomial runtime
> > > 3. typo: The axies* are indices of A;
> > > 4. typo: th3, positive integer, and donote*
> >
> > **Response:** We thank you for pointing out these typos. We have fixed these typos in our revised manuscript on pages 4, 14, 2, and 6.

---

### Official Review · Reviewer_rGCR · 2024-11-04

**Soundness:** 3
**Presentation:** 3
**Contribution:** 3
**Rating:** 6
**Confidence:** 3

**Summary:**

The paper presents a novel framework for approximating Sparse PCA by decomposing the original large-scale problem into smaller subproblems through matrix block-diagonalization. The method involves three key steps: (1) transforming the input covariance matrix into a block-diagonal form by thresholding and grouping non-zero entries, (2) applying any existing Sparse PCA algorithm to each block independently, and (3) reconstructing the overall solution from the subproblem results. This approach aims to achieve significant computational speedups while maintaining high solution quality, supported by theoretical guarantees and extensive empirical evaluations on various large-scale datasets.

**Strengths:**

1. The framework significantly reduces runtime by decomposing the original problem into smaller, more manageable subproblems.

2. The paper provides approximation guarantees and time complexity analyses, ensuring that the method preserves solution quality.

3. Extensive experiments on diverse, large-scale datasets demonstrate the framework’s effectiveness in reducing computational time with minimal approximation errors.

**Weaknesses:**

1. The core idea of decomposing a large-scale optimization problem into smaller subproblems and then combining their solutions has been previously explored in the literature (e.g., [1,2,3]). The paper does not sufficiently acknowledge or discuss these existing approaches, missing an opportunity to position its contribution within the broader context of optimization techniques.

2. While each subproblem is solved under a $k$-sparsity constraint, it is not explicitly clear how the framework ensures that the combined solution across all subproblems maintains the overall $k$-sparsity required by the Sparse PCA problem. This raises concerns about the validity of the final solution's sparsity, which is crucial for interpretability and efficiency.

[1] "Exact covariance thresholding into connected components for large-scale graphical lasso." Journal of Machine Learning Research, 13(27):781−794, 2012.

[2] "Graphical lasso and thresholding: Equivalence and closed-form solutions." Journal of Machine Learning Research, 20(10):1−44, 2019.

[3] "Learning large-scale MTP2 Gaussian graphical models via bridge-block decomposition." Advances in Neural Information Processing Systems 36:73211-73231, 2023.

**Questions:**

Please address the concerns outlined in Weaknesses.

---

> ### Author Response · Authors · 2024-11-22
>
> Dear Reviewer,
>
> We greatly appreciate your valuable feedback and constructive suggestions. Below, we have provided our responses and clarifications. We believe we have addressed your concerns and would be grateful if you could consider increasing your scores.
>
> **Responses to weaknesses**:
>
> 1. > The core idea of decomposing a large-scale optimization problem into smaller subproblems and then combining their solutions has been previously explored in the literature (e.g., [1,2,3]). The paper does not sufficiently acknowledge or discuss these existing approaches, missing an opportunity to position its contribution within the broader context of optimization techniques.
>
> **Response:** Thank you for your suggestion. We have added additional discussion on this point and position our contribution to the broader context of optimization techniques, in Appendex A on Page 14 in our revised manuscript.
>
> 2. > While each subproblem is solved under a $k$-sparsity constraint, it is not explicitly clear how the framework ensures that the combined solution across all subproblems maintains the overall $k$-sparsity required by the Sparse PCA problem. This raises concerns about the validity of the final solution's sparsity, which is crucial for interpretability and efficiency.
>
> **Response:** We appreciate it for raising this important point. Our framework (Algorithm 3) actually ensures that the final solution remains $k$-sparse, as long as it calls a valid SPCA algorithm $\mathcal{A}$. Indeed, the final output is guaranteed to be $k$-sparse thanks to the construction on lines 5--8 of Algorithm 3, provided each $y_i$ is $k$-sparse. Examples of such SPCA algorithms $\mathcal{A}$ include Chan's algorithm or Branch-and-Bound algorithm, both of which are discussed in detail in our paper.

---

### Meta-Review · Area_Chair_C4f6 · 2024-12-22

**Metareview:**

This paper presents an efficient approach to Sparse Principal Component Analysis (SPCA) by leveraging a block-diagonal approximation of the covariance matrix, thereby reducing the original problem to a series of smaller subproblems. The paper is well-written, with the key ideas and contributions clearly presented. The authors provide  theoretical results, and the experimental section includes comparisons with other state-of-the-art methods on a variety of datasets, demonstrating advantages in both runtime efficiency and average approximation error. The reviewers acknowledged the contributions of this work, and the authors sufficiently addressed the minor concerns raised during the rebuttal phase. Overall, this paper makes a clear contribution to scalability of sparse PCA, which is a relevant topic of research.  Therefore, I recommend its acceptance to ICLR.

**Additional Comments On Reviewer Discussion:**

During the rebuttal the authors addressed concerns of reviewer's rGCR  regarding an issue with the positioning of this paper within the broader context and k-sparsity of the final solution. Moreover, they provided clarifications to several issues raised by reviewer ZrrB on significance of the theoretical contributions and lack of comparison with existing methods. Reviewer ZrrB acknowledged that the responses addressed to a large extend their concerns. The authors also provided detailed responses to all concerns of uKtp and fTk5 who expressed some concerns regarding presentation of the main contributions. Reviewer fTk5 found the responses convincing and raised the score.

---

### Decision · Program_Chairs · 2025-01-22

Accept (Poster)